# Control of lysogeny and antiphage defense by a prophage-encoded kinase-phosphatase module

Yunxue Guo[1,2,3,10], Kaihao Tang [1,2,10], Brandon Sit [4,5,9,10], Jiayu Gu[1,3,10], Ran Chen[1,2], Xinqi Shao[6], Shituan Lin[1,3], Zixian Huang[1,3], Zhaolong Nie[1], Jianzhong Lin[1,3], Xiaoxiao Liu[1,2], Weiquan Wang[1,3], Xinyu Gao[1,3], Tianlang Liu[1,3], Fei Liu[6], Hongbo R. Luo [7], Matthew K. Waldor [4,5,8] ✉ & Xiaoxue Wang [1,2,3] ✉

The filamentous 'Pf' bacteriophages of *Pseudomonas aeruginosa* play roles in biofilm formation and virulence, but mechanisms governing Pf prophage activation in biofilms are unclear. Here, we identify a prophage regulatory module, KKP (kinase-kinase-phosphatase), that controls virion production of co-resident Pf prophages and mediates host defense against diverse lytic phages. KKP consists of Ser/Thr kinases PfkA and PfkB, and phosphatase PfpC. The kinases have multiple host targets, one of which is MvaU, a host nucleoid-binding protein and known prophage-silencing factor. Characterization of KKP deletion and overexpression strains with transcriptional, protein-level and prophage-based approaches indicates that shifts in the balance between kinase and phosphatase activities regulate phage production by controlling MvaU phosphorylation. In addition, KKP acts as a tripartite toxin-antitoxin system that provides defense against some lytic phages. A conserved lytic phage replication protein inhibits the KKP phosphatase PfpC, stimulating toxic kinase activity and blocking lytic phage production. Thus, KKP represents a phosphorylation-based mechanism for prophage regulation and antiphage defense. The conservation of KKP gene clusters in >1000 diverse temperate prophages suggests that integrated control of temperate and lytic phage infection by KKP-like regulatory modules may play a widespread role in shaping host cell physiology.

Filamentous bacteriophages (inoviruses) infect diverse bacterial hosts[1,2], including the opportunistic human pathogen *Pseudomonas aeruginosa*[3]. The filamentous 'Pf' phages in *P. aeruginosa* play important roles in virulence[4,5] and formation of characteristic *P. aeruginosa* biofilms[6,7], where genes enabling virion production are markedly upregulated[8]. Pf virions participate in the formation of viscous liquid crystals that support biofilm structure[9–11] and can suppress mammalian immunity by directly inhibiting phagocytosis and bacterial clearance[12]. Several host factors, such as H-NS-like nucleoid binding proteins, regulate the lysogeny of the Pf4 prophage[13–16], but the mechanisms that

control Pf prophage activation, especially in infection-relevant conditions such as biofilms, are largely uncharted. Pfs often encode conserved structural genes as well as disparate accessory genes whose functions are not clear[1,17,18], suggesting genetic diversity within the Pf clade may have consequences for understanding Pf prophage biology and regulation. Additionally, although many *P. aeruginosa* strains encode multiple Pf prophages, interactions between co-resident Pf prophages have not been documented.

In this work, we identified the dual function of a filamentous 'Pf' prophage encoded tripartite toxin-antitoxin system KKP

(kinase-kinase-phosphatase) in controlling virion production of co-resident Pf prophages and mediating host defense against diverse lytic phages. The balance between toxin KK (consists of Ser/Thr kinases PfkA and PfkB) and antitoxin phosphatase PfpC activities regulate co-resident Pf phage production by controlling the phosphorylation of a known prophage silencer MvaU. The KKP modules from different temperate phages provide defense against diverse lytic phages. A conserved lytic phage replication protein inhibits the KKP phosphatase PfpC, stimulating toxic kinase activity and blocking lytic phage production.

## Results

### Divergent Pf production in two globally-distributed PAO1 lineages

During studies of *P. aeruginosa* biofilm formation in a flow cell-based catheter system (Fig. 1a) that mimics chronic infection[19], we observed that two major lineages of the prototypical *P. aeruginosa* strain PAO1 produced dramatically different titers of Pf virions (Fig. 1b). These lineages are designated 'PAO1', the original strain isolated in 1955[20], and 'MPAO1', a widely-distributed PAO1 derivative[21]. In the flow cell system, PAO1 and MPAO1 both formed visible thick, crystal violet-positive biofilms on the inner surface of the catheter after 3 days of growth (Fig. 1a). However, biofilm effluents from the two strains contained marked differences in phage titers. MPAO1, but not PAO1, produced a high amount of Pf phage, reaching ~10^10 PFU/ml after 6 days (Fig. 1b). TEM of MPAO1 effluents at day 6 revealed characteristic bundles of phages with a wavy ~2–3 μm long filamentous structure (Fig. 1b). To visualize in situ Pf production in biofilms, the major coat protein (pVIII) of the Pf4 prophage, which is present in both MPAO1 and PAO1, was fused in-frame with GFP, creating a reporter of Pf4 production (Fig. 1c). Consistent with the PFU and TEM data, Pf4 production was readily observed in ridge-like structures within MPAO1 biofilms, but was only rarely detected in PAO1 biofilms (Fig. 1c).

Prior comparative genomics of PAO1 and MPAO1 has revealed minor genetic differences between these *P. aeruginosa* sublines that likely impact important phenotypes such as pyocyanin and pyoverdine production and virulence in animal models[22,23]. The largest difference between MPAO1 and PAO1 is the presence of a second Pf prophage, Pf6, in MPAO1. Whole-genome sequencing confirmed that PAO1 only harbors the Pf4 prophage, whereas MPAO1 harbors both the Pf4 and Pf6 prophages (Fig. 1d; Supplementary Table 1). The Pf4 prophages in MPAO1 and in PAO1 are nearly identical, differing only in an 18 bp deletion near the *attL* site in MPAO1. The Pf6 prophage (12,152 bp) in MPAO1 is inserted in the tRNA^Met gene, and the core region of the prophage, which encodes the virion capsid, assembly, and replication-related proteins (PA0717-RepF6), is highly similar (>98% nucleotide identity) to the core region of Pf4[23]. Since MPAO1 harbors two Pf prophages, to test which prophage contributes to Pf virion production in biofilms, we generated MPAO1 strains lacking the entirety of Pf4 (MP-ΔPf4), the Pf6 core region (MP-ΔPf6core), or both (MP-ΔPf4ΔPf6core). All three mutant strains efficiently formed biofilms, and the deletion of either or both prophages did not affect planktonic cells growth (Supplementary Fig. 1a, b). Phage titers in biofilm effluents from wild-type (WT) MPAO1 and the two single mutant strains were similar (Fig. 1e, Supplementary Fig. 1a), indicating that Pf4 and Pf6 can be produced independently of each other's core regions. We validated this result with quantitative PCR (qPCR) of Pf4 (*pfiA*) or Pf6 (*intF6*)-specific loci in biofilm effluents (Fig. 1f). No virions were detected in the biofilm effluent of MP-ΔPf4ΔPf6core (Fig. 1e).

Pf4 production in biofilms has been associated with elevated *P. aeruginosa* resistance to aminoglycoside antibiotics[10]. Biofilms from all three phage-producing strains (MPAO1, MP-ΔPf4, and MP-ΔPf6core) exhibited markedly higher resistance to gentamicin (~30% survival) compared to the non-producer (MP-ΔPf4ΔPf6core (<10^−7% survival)) (Fig. 1g), suggesting that production of Pf6 in MPAO1 biofilms also promotes antibiotic resistance similarly to Pf4. These phenotypes were dependent on biofilm formation, as mechanical disruption of biofilms derived from the three phage-producing strains markedly diminished their resistance to gentamicin (Fig. 1g). Importantly, PAO1 biofilms, which are deficient for Pf production (Fig. 1b), and planktonic cells of all four MPAO1 strains were highly sensitive to gentamicin, demonstrating that virion production and biofilm structure are co-requisites of reduced antibiotic susceptibility in MPAO1.

Pf virion production has also been linked with *P. aeruginosa* virulence[4], and the pathogen is known to form biofilms in vivo[24,25]. In mouse models of intratracheal *P. aeruginosa* infection using planktonic or biofilm-derived inocula, deletion of both Pf prophages greatly reduced MPAO1-induced lethality (Fig. 1h). Together, these data reveal that both Pf4 and Pf6 prophages give rise to Pf virions in MPAO1 biofilms, and that either (or both) of these Pfs modulate host cell phenotypes in biofilm conditions and *P. aeruginosa* virulence. Literature-based analyses of academic networks revealed that while many groups currently use MPAO1 (often simply referred to as 'PAO1'), use of Pf6-deficient PAO1 continues, suggesting that caution be taken in interpreting data involving *P. aeruginosa* strains with varying Pf prophage carriage (Supplementary Table 2 and Supplementary Data 1).

### The Pf6 KKP_{MP} module controls Pf lysogeny via MvaU phosphorylation

Since Pf4 production was maintained in MP-ΔPf6core but absent in PAO1 (which completely lacks Pf6), we investigated how Pf6 controls Pf4 production. Initially, RNA-seq studies were used to probe the pathways that control Pf4 and Pf6 prophage activation in MPAO1 biofilms. The Pf4 and Pf6 core genes, including their respective excision (Pf4: *xisF4*, Pf6: *xisF6*), replication (Pf4: *repF4*, Pf6: *repF6*), and major coat protein genes (*gVIII*) were among the most upregulated genes in MPAO1 biofilm cells versus planktonic cells, where no Pf phage particles were detected (Fig. 2a, Supplementary Table 3). In contrast, no Pf4 prophage genes were significantly induced in PAO1 biofilm cells. Thus, the presence of the two related prophages in MPAO1 dramatically shifts prophage gene expression in biofilms towards virion production.

Given the marked similarity of the Pf4 and Pf6 core regions, we hypothesized that the three accessory genes present in Pf6, but not Pf4, are important for Pf prophage induction in biofilms. Interestingly, two of these loci encode proteins with strong similarity to eukaryotic-like serine/threonine kinases (PF00069), and the third to a serine/threonine phosphatase (PF13672) (Supplementary Fig. 2). We refer to these three loci as *pfkA*, *pfkB* and *pfpC*, and the entire operon as KKP_{MP} (for kinase-kinase-phosphatase in MPAO1) (Fig. 2b). Coupling temporal gene expression analyses with measurements of Pf titers revealed that the ratio of kinase to phosphatase gene expression ((*pfkA*+*pfkB*)/*pfpC*) was inversely correlated with prophage gene expression and virion production (Fig. 2c, Supplementary Fig. 3a, b, Spearman's $r = -0.9286$, $P = 0.0022$). In planktonic cells, where Pf virions are not produced, *pfkA* was the most highly expressed KKP_{MP} gene (Fig. 2b, Supplementary Fig. 3a, b, Supplementary Table 3). In biofilms, relative phosphatase (*pfpC*) gene expression increased as Pf production rose during the first 4-5 days of biofilm formation, reaching peak titers by day 6–8 (Fig. 2c, Supplementary Fig. 3a). These data suggest that the KKP_{MP} kinases play a role in maintaining prophage quiescence. In support of this hypothesis, targeted deletion of *pfkA* from MPAO1 led to accelerated and heightened production of Pf4 and Pf6 virions in biofilms, where high titers of Pf phages were already produced by day 3 (Fig. 2d). While the virus-to-microbe (VMR) ratio remained ~2:1 in WT MPAO1 biofilms, this ratio increased in developing MPAO1 Δ*pfkA* biofilms up to a maximum value of 28:1 (Fig. 2e). Similar increases in Pf virion production and VMR were also observed in MPAO1 ΔKK_{MP}, which lacks both kinases (Fig. 2d, e). Thus, the KKP-encoded kinases appear to play a critical role in linking Pf4 and Pf6 development to

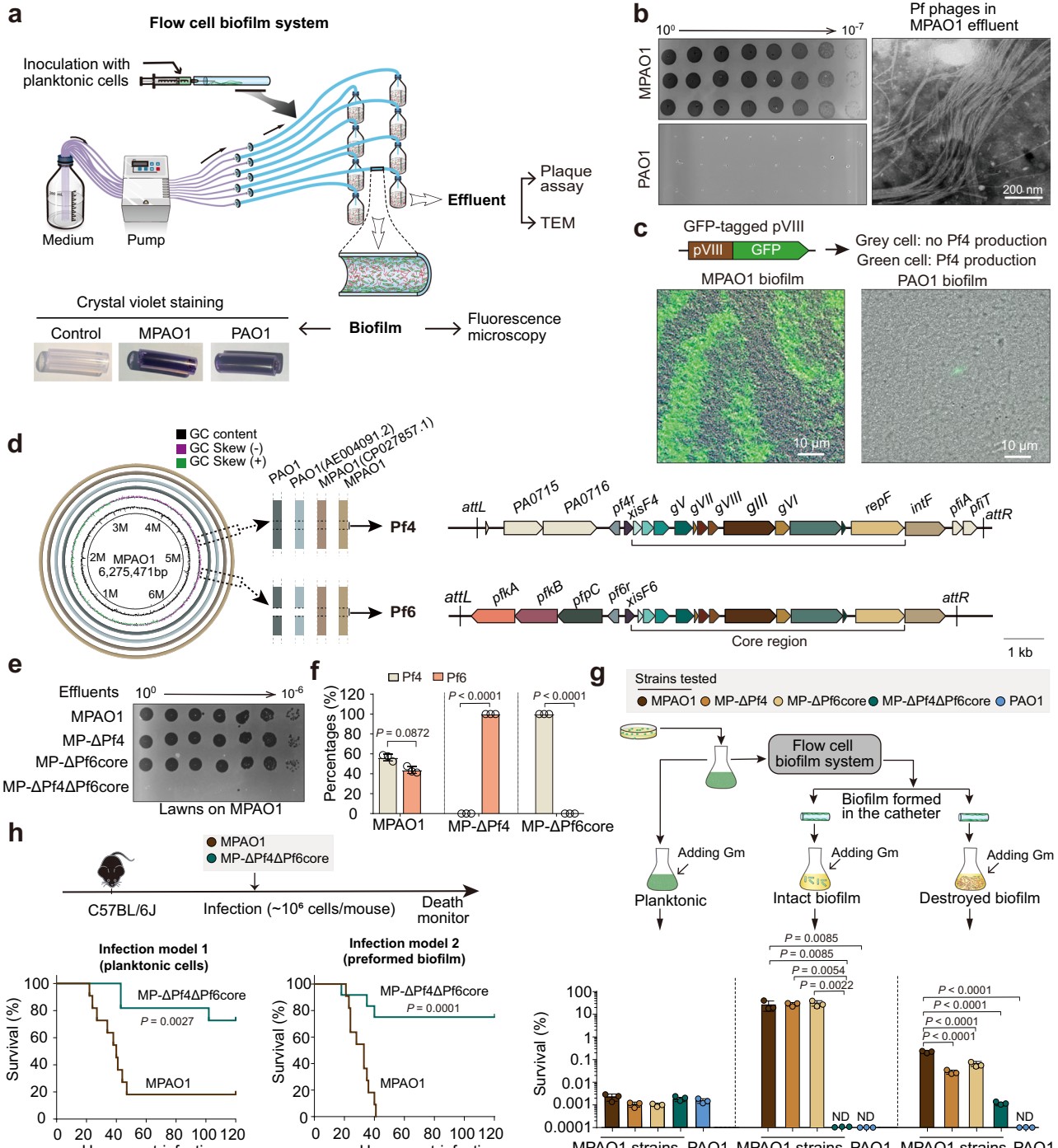

**Fig. 1 | Pf4 and Pf6 are coordinately produced in MPAO1 biofilms and contribute to host antibiotic resistance and virulence. a** Schematic of the flow cell biofilm system used in this study. Images in bottom left show representative crystal violet-stained catheter sections (purple staining indicates biofilm formation). Planktonic cells are free-living cells grown in liquid culture. **b** Phage titers (left) and transmission electron microscopy (TEM, right) of biofilm effluents from day 6 MPAO1 biofilms. **c** Fluorescence microscopy of day 6 biofilms in MPAO1 (left) or PAO1 (right) strains expressing GFP-tagged Pf4. **d** Genome comparisons of de novo sequenced MPAO1, PAO1, and the first NCBI-deposited MPAO1 and PAO1 genomes. MPAO1 sequenced in this study was used to annotate the Pf4 and Pf6 prophages (shown at right with core Pf regions marked). **e** Representative phage titers of day 6 biofilm effluents from the indicated MPAO1 derivatives plated on lawns of the MPAO1 strain. **f** Percentages of Pf4 and Pf6 phages in biofilm effluents from the MPAO1 WT and mutant strains quantified by qPCR. Data are shown as the mean ± SD. Two-sided Student's *t* Test was used for comparisons between Pf4 and Pf6 in each strain ($n = 3$). $P < 0.05$ was considered statistically significant. **g** Gentamicin (Gm) sensitivity of various strains in varying biofilm states. Experimental design is shown in top panel. ND: not detected. Data are shown as the mean ± SD. Ordinary one-way ANOVA with Tukey's multiple comparisons test was used ($n = 3$), and $P < 0.05$ was considered statistically significant. **h** Survival of C57BL/6J mice infected with planktonic or biofilm-state MPAO1 *P. aeruginosa* WT ($n = 11$) and ΔPf4ΔPf6core ($n = 10$ and 12 for planktonic and biofilm-state, respectively). The log-rank (Mantel-Cox) test with 95% confidence interval was used, and $P < 0.05$ was considered statistically significant.

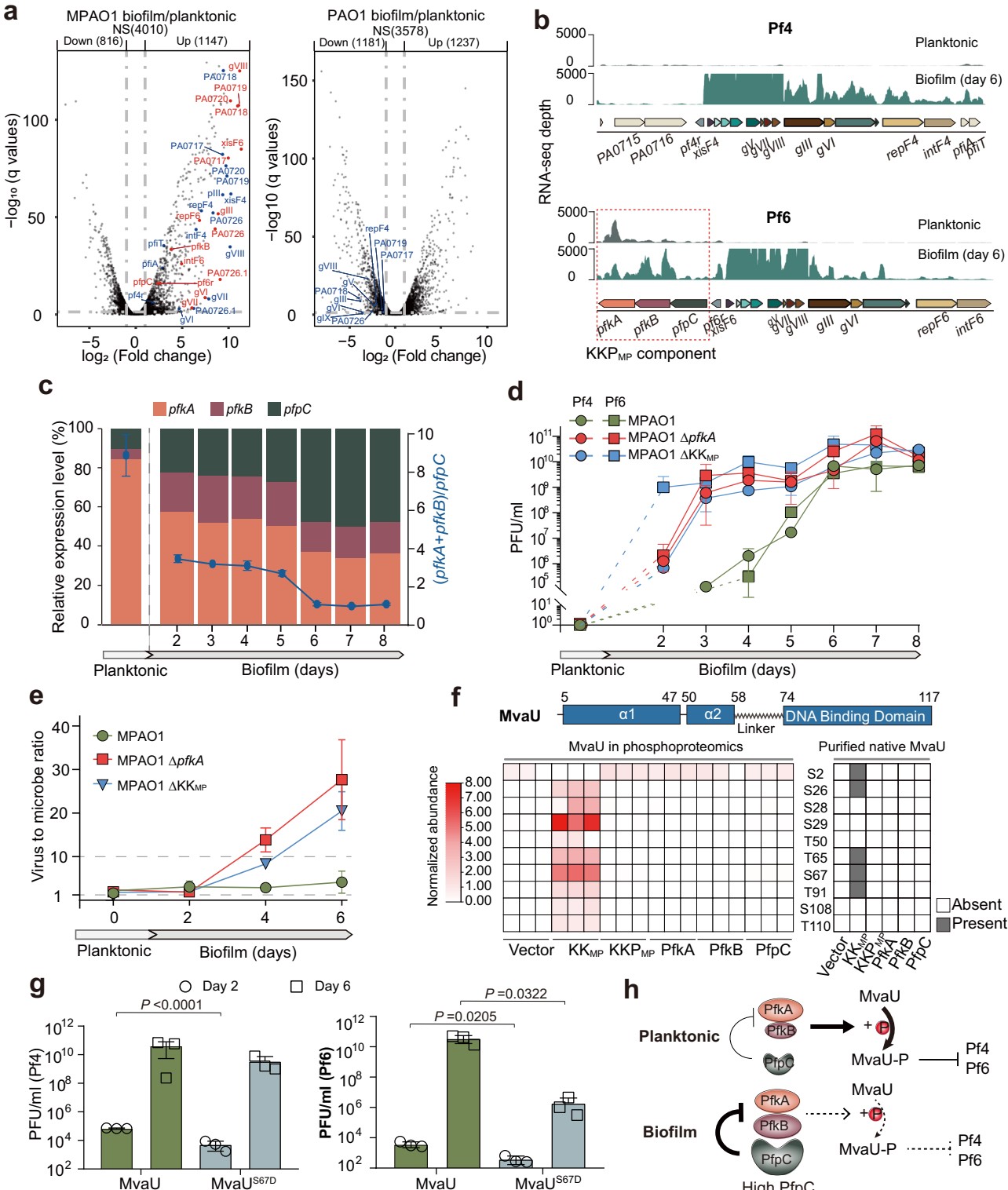

environmental conditions, as their reduced expression relative to *pfpC* in biofilms permits controlled phage induction.

We hypothesized a phosphorylated substrate of PfkA and/or PfkB could enable KKP$_{MP}$ regulation of Pf gene activation. Phosphoproteomics of MPAO1 cells overexpressing PfkA identified several putative host Ser/Thr phosphorylation substrates, such as DnaG, Rho, and the H-NS-like protein MvaU (Supplementary Table 4). We were particularly intrigued by the identification of MvaU, since this and the related H-NS-like factor MvaT are known to coordinately silence Pf production in *P.*

*aeruginosa*, and their combined deletion has been reported to enable Pf production in PAO1 planktonic cells[13,15]. To further dissect MvaU phosphorylation without interference from natively-expressed KKP$_{MP}$ or Pf6, we conducted additional phosphoproteomics in PAO1 cells that expressed each KKP$_{MP}$ component separately or in combination. No or very low levels of MvaU phosphorylation were detected when only one kinase was expressed in PAO1. In contrast, expression of both PfkA and PfkB led to high phosphorylation of MvaU, suggesting the presence of both kinases was required for substrate phosphorylation (Fig. 2f). The

**Fig. 2 | The Pf6 KKP$_{MP}$ module controls Pf lysogeny via MvaU phosphorylation. a** Differentially-expressed genes in biofilms versus planktonic cells in MPAO1 and PAO1. Statistically significantly changed Pf6 and Pf4 genes are labeled in red and blue, respectively. NS, not significant. **b** RNA-seq read coverage of Pf4 and Pf6 transcripts in MPAO1 biofilm and planktonic cells. Pf6 KKP$_{MP}$ genes (*pfkA*, *pfkB* and *pfpC*) are boxed. **c** Relative mRNA expression levels of *pfkA*, *pfkB* and *pfpC* in MPAO1 planktonic and biofilm cells (day 2–8). The (*pfkA+pfkB*)/*pfpC* ratio is shown as a blue line. **d** Pf phage titer kinetics in biofilms from MPAO1, MPAO1 Δ*pfkA*, and MPAO1 ΔKK$_{MP}$ using methods established in Supplementary Fig. 3c–e. **e** Virus to microbe ratios (VMR) in biofilms of MPAO1, MPAO1 Δ*pfkA* and MPAO1 ΔKK$_{MP}$ were determined by qPCR using primers to amplify an identical fragment of Pf4 and Pf6 and to amplify chromosomal gene *gyrB*. **f** Top: MvaU domain schematic. Left: Phosphorylation sites in native MvaU identified by phosphoproteomics in PAO1 with expression of indicated KKP$_{MP}$ components. Right: Phosphorylation sites in native MvaU-His purified from MPAO1 with indicated KKP$_{MP}$ components. **g** Effects of chromosomal expression of MvaU WT or MvaU$^{S67D}$ on the production of Pf4 and Pf6 during biofilm formation at day 2 and day 6. Data are shown as the mean ± SD. Two-sided Student's *t* Test was used for comparisons of phage titers between MvaU and MvaU$^{S67D}$ at each day ($n = 3$), and $P < 0.05$ was considered statistically significant. **h** Model for KKP$_{MP}$ control of Pf lysogeny via MvaU phosphorylation. In planktonic (free-living) cells, where there is relatively minimal expression of PfpC, PfkA and PfkB lead to MvaU phosphorylation and maintain Pf lysogeny. In biofilms, elevated PfpC expression inhibits kinase activity, leading to MvaU dephosphorylation, elevated Pf gene expression and phage propagation. The size of each labeled shape corresponds to relative expression levels for each protein in each condition. At least two independent cultures were used ($n = 3$ for **a**, **c**, **d**, **f**, **g**; $n = 2$ for **e**), and data are shown as the mean ± SD in (**c–e**, **g**).

expression of intact KKP$_{MP}$ entirely inhibited MvaU phosphorylation, indicating that PfpC counteracts PfkA and PfkB activity. Similar results were obtained with targeted mass spectrometry of chromosomally-expressed His-tagged MvaU purified from MPAO1 cells overexpressing KKP$_{MP}$ components, where we observed that 0.5–20% of His-MvaU was phosphorylated (Fig. 2f, Supplementary Fig. 4a, b). Thus, the host H-NS protein MvaU, a known regulator of Pf lysogeny, is a target of KKP$_{MP}$ phosphorylation.

To probe whether MvaU phosphorylation influences Pf lysogeny, we individually substituted selected MvaU residues identified by phosphoproteomics as phosphorylated by KK$_{MP}$ (S26, T50, T65, S67, S108) with either aspartate (phosphomimetic) or alanine (nonphosphorylatable) residues (Fig. 2f). We initially evaluated the consequences of these substitutions in Δ*mvaU*Δ*mvaT* cells, which produce Pf virions in planktonic cells. In this background, expression of MvaU$^{S67D}$ totally repressed Pf virion production and expression of Pf4 and Pf6 genes, but the corresponding Ala (A) mutation had no effect (Supplementary Fig. 4c, d). These data suggest that phosphorylation of MvaU by the KK$_{MP}$ kinases promote its capacity to inhibit Pf gene expression. Interestingly, co-expression of MvaU$^{S67D}$ and MvaU$^{S67A}$ still efficiently suppressed Pf activation, indicating that partial phosphorylation of the cellular MvaU pool is sufficient for silencing Pf (Supplementary Fig. 4c). To test the impact of MvaU$^{S67D}$ on Pf production in biofilms, we compared Pf production in MPAO1 harboring chromosomal *mvaU* WT or *mvaU$^{S67D}$*. Both Pf4 and Pf6 phage production were reduced in the strain expressing MvaU$^{S67D}$ in biofilms (Fig. 2g). Collectively, these results suggest a model where the relatively high expression of PfkA and PfkB in planktonic MPAO1 cells leads to MvaU phosphorylation and downstream silencing of Pf prophage gene expression. On entry to the biofilm state, elevated PfpC expression leads to the dephosphorylation of MvaU, which in turn relieves repression of prophage genes that promote Pf phage production in MPAO1 biofilms (Fig. 2h).

## KKP gene clusters are widespread in diverse prophages
To investigate whether KKP modules are conserved outside of *P. aeruginosa*, we performed bioinformatic searches to identify triplet gene combinations with the same gene organization as KKP$_{MP}$ (i.e., kinase-kinase phosphatase). We only included KKP clusters where the specific domain organization of the two kinases and phosphatase was also conserved. PfkA and PfkB share an N-terminal kinase domain, but they have divergent C-terminal domains. PfkB has a forkhead-associated domain (FHA, PF00498) that is a phosphopeptide interaction motif present in both eukaryotes and prokaryotes[26] (Fig. 3a, Supplementary Fig. 2), whereas PfkA has a cold shock domain (CSD, PF00313) with a conserved GFI/A motif, and the CSD domain may have ssDNA/RNA-binding properties[27]. Over 1200 putative KKP gene clusters were identified with this approach, accounting for 91.1% of all observed gene triplets with a phosphatase followed by two kinases with variable C-termini (Fig. 3b). These clusters were restricted to

Gram-negative species, and primarily found in Proteobacteria, but also in Cyanobacteria and Bacteroidetes (Supplementary Datas 2, 3). Interestingly, PfkA-like kinases fused to a CSD were only present in prokaryotes, suggesting PfkA may be a defining feature of KKP modules (Fig. 3c, d).

Phylogenetic analysis using PfkA sequences revealed that KKP gene clusters can be broadly classed into three clades (Fig. 3e). KKP modules from different clades were found in the same bacterial species; e.g., clade I and II modules were identified in different *E. coli* isolates. Highly similar KKP gene clusters (>90% amino acid identity) were found in gut-resident organisms like *Enterobacter*, *Escherichia* and *Salmonella* (Fig. 3e), suggesting that KKP gene clusters can be acquired through horizontal gene transfer. In >90% of cases, KKP gene clusters were located in the genomes of diverse temperate phages including P2 and its satellite P4, Mu-like, and Pf6-like prophages (Fig. 3f, g, Supplementary Data 2). Similar to MPAO1, KKP clusters were found in several strains with multiple potentially interacting prophages. For example, *Shewanella* sp. W3-18-1 harbors a P4 satellite prophage containing a KKP cluster along with a P2-family prophage that may correspond to its helper. Outside of prophages, KKP clusters were frequently found closely linked to genes associated with mobile genetic elements (70 out of 114), including integrases and transposons, as well as modules such as TA and restriction-modification systems (Fig. 3f, g, Supplementary Data 2) that are associated with bacterial defense against mobile elements.

## KKP is a phosphorylation-based toxin/antitoxin system
The opposing kinase and phosphatase functions encoded by KKP$_{MP}$ prompted us to ask if this gene cluster constitutes a TA system. Expression of each of the three genes alone or all of the genes together in PAO1 did not affect growth (Fig. 4a, Supplementary Fig. 5a). However, the co-expression of PfkA and PfkB almost completely inhibited growth, suggesting that the kinases jointly exert toxic activity (Fig. 4a, Supplementary Fig. 5a). The catalytic motifs in PfpC's N-terminal phosphatase domain were required for counteracting PfkA/B-mediated toxicity (Supplementary Fig. 5b), strongly suggesting that PfpC neutralizes PfkA and PfkB toxicity through dephosphorylation. Thus, KKP$_{MP}$ functions as a tripartite TA system where the two kinases together function as the toxin and the phosphatase as the antitoxin.

We next generated a series of PfkA and PfkB mutants and determined their impact on growth. Both of the PfkA and PfkB Ser/Thr kinase domains have conserved GGM, HRD, and DFG motifs at their respective N-termini (Fig. 4a). Single or multiple-motif mutations in either kinase domain had no effect on cell growth, whereas mutations in the HRD or DFG motifs in both kinases abolished toxicity (Fig. 4a, Supplementary Fig. 6a), suggesting the two kinase domains have similar but non-redundant activity. An additional set of C-terminal truncation mutants and GFI motif mutation revealed that both the PfkA CSD and PfkB FHA domains are critical for their joint toxicity

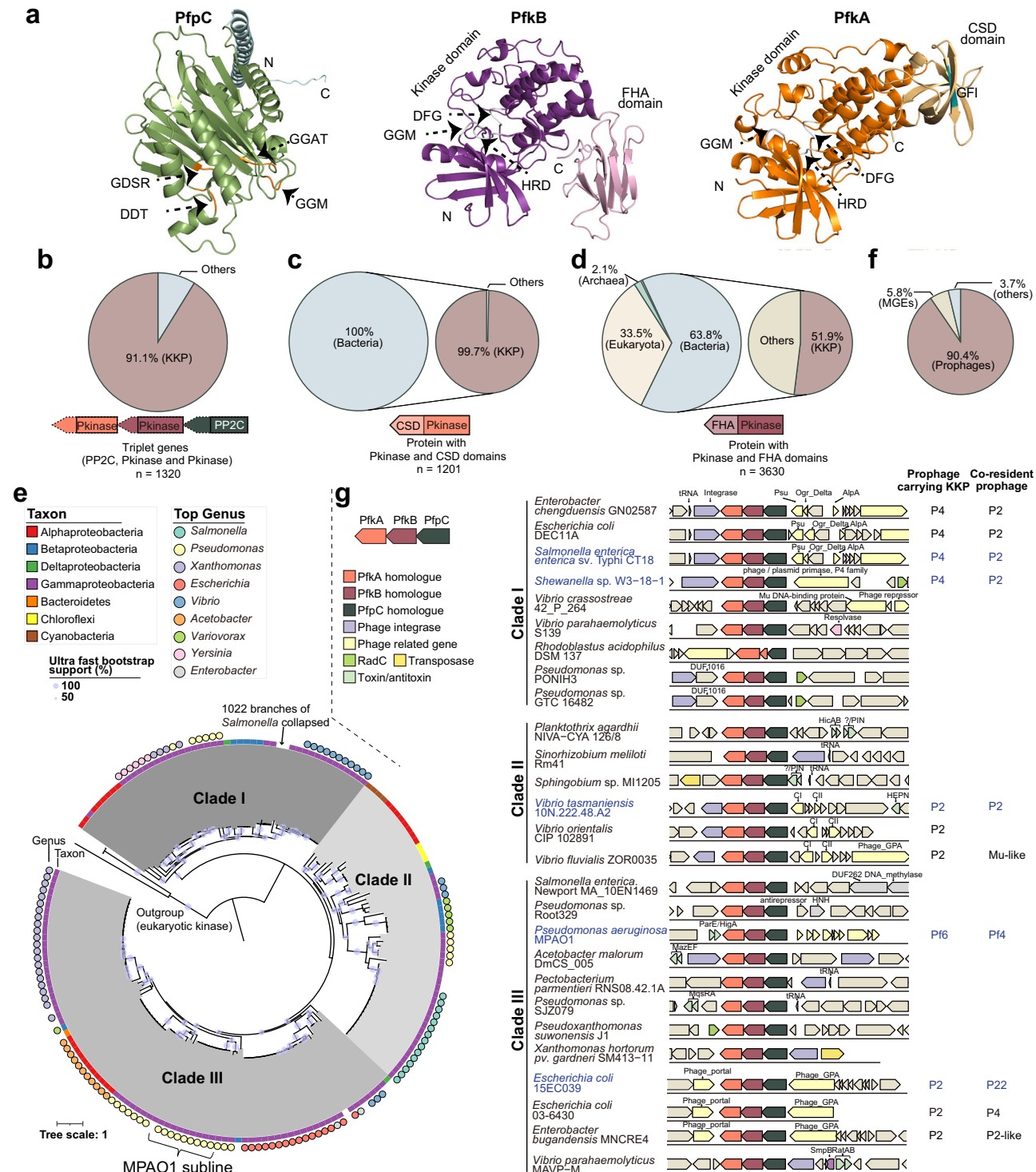

**Fig. 3 | KKP modules are widespread in diverse Gram-negative organisms.**
**a** Predicted structures of KKP_MP components by AlphaFold2. Conserved phosphatase motifs are shown in orange for PfpC. Conserved kinase motifs are shown in gray for PfkB and PfkA. **b** Distribution of triplet genes with a phosphatase domain (PF13672, PP2C) followed by two kinase domains (PF00069, Pkinase) before further filtering by CSD and FHA domain status. **c** Distribution of kinases (PF00069: Pkinase) fused with a CSD domain (PF00313: CSD). Left: taxonomic distribution. Right: proportion of bacterial kinases from left pie chart located in authentic KKP gene clusters. **d** Distribution of kinases (PF00069: Pkinase) fused with an FHA domain (PF00498: FHA). Left: taxonomic distribution. Right: proportion of bacterial kinases from left pie chart located in authentic KKP gene clusters. **e** Maximum likelihood tree of PfkA-like kinases. Colored bars around the tree indicate taxa. Filled circles indicate specific genera. Note that many *Salmonella* sequences were

collapsed. **f** Genomic location distribution of KKP modules. **g** Representative KKP gene clusters with their genomic neighborhoods. KKP modules located in prophages and their co-resident prophages are indicated on the right, and other KKP-linked genes (transposons, TA systems, and restrictive modification systems) are indicated on the left. Gene names or conserved domains shown include: Psu, phage polarity suppression protein (PF07455); Phage_GPA, Bacteriophage replication gene A protein (GPA, PF05840); DNA_methylase, C-5 cytosine-specific DNA methylase (PF00145); RadC, DNA repair protein RadC (PF04002); DUF262 (PF03235); HNH, HNH endonuclease (PF01844); Ogr_Delta, viral family of phage zinc-binding transcriptional activators (PF04606); AplA, transcriptional regulator, AlpA family (PF05930); Phage_portal, Phage portal protein (PF04860); DUF1016, YhcG PDDEXK nuclease domain (PF06250); CI and CII, Bacteriophage repressor. The KKPs in prophages that were selected for follow-up are labeled in blue.

(Supplementary Fig. 6a, b). Reciprocal pulldowns in an *E. coli* heterologous expression system along with structural modeling additionally suggested that PfkA and PfkB interact through their respective N and C-terminal domains, consistent with the idea that the two kinases function together to mediate toxicity (Fig. 4b, Supplementary Fig. 6c, Supplementary Table 5–10).

To understand the mechanism of kinase-mediated toxicity, we performed untargeted phosphoproteomics in PAO1 *P. aeruginosa* expressing PfkA and/or PfkB. While expression of neither kinase by itself led to appreciable phosphorylation of host targets, PfkA and PfkB co-expression resulted in phosphorylation of many cellular proteins (Fig. 4c). Many putative PfkA/B targets are involved in essential cellular functions, including DNA replication (PolA, TopA, DnaG), translation (InfB, Tsf), and cell division (FtsZ, MinCE) (Fig. 4c, Supplementary Data 4). The phosphoproteomics dataset also revealed multiple phosphorylated sites in PfkA (T34, S37, T271) and PfkB (T261, T319), which we confirmed by mass spectrometry of kinases purified from PfkA/B co-expressing cells (Supplementary Fig. 6d). Phosphorylation of PfkA/B themselves in single kinase-expressing strains was very low (Supplementary Fig. 6d), indicating that akin to host targets, autophosphorylation, which is a hallmark of eukaryotic-type Ser/Thr kinase activation[28], requires the presence of both kinases. Both host and PfkA/B phosphorylation were sharply reduced when PfpC was also present, underscoring the idea that PfpC is a phosphatase antitoxin (Supplementary Fig. 6d).

Since phosphorylation is a dynamic and potentially reversible post-translational modification, we sought to understand how the duration or strength of kinase activity impacted phenotypic consequences of KKP activation. In PAO1, prolonged co-expression of *pfkA* and *pfkB* (KK$_{MP}$), but not KKP$_{MP}$, led to cell filamentation and loss of membrane integrity (Fig. 4d, Supplementary Fig. 7a, b). The filamentous phenotype is consistent with the identification of cell division-related proteins such as FtsZ and MinC/E as PfkA/B targets (Supplementary Table 5). Expression of PfkA/B or PfpC from independent plasmids in *E. coli* revealed that short periods of kinase induction led to apparent delays in cell division that were reversed by subsequent induction of PfpC (Fig. 4e). Using a modified two-plasmid system to independently control PfkA and PfkB expression, we found that joint kinase toxicity is dose-dependent and requires relatively high expression of PfkB (Fig. 4f). The necessity for different levels of expression of PfkA and PfkB for toxicity provide a plausible explanation for distinct phenotypic outputs of the kinases in different conditions. In normally growing cells, PfkB expression is sufficient to support PfkA activity, but too low to engender toxicity (Fig. 2b). In contrast, when levels of PfkB increase without corresponding increases in PfpC, kinase-based toxicity occurs (Fig. 4e).

## KKP provides defense against phage infection through kinase toxicity

In addition to regulating Pf activation, we asked whether KKP$_{MP}$ had a role in phage defense, as TA systems in other bacteria have recently been linked to bacterial defense against lytic phages[29–32]. Remarkably, introduction of KKP$_{MP}$ into PAO1 conferred strong protection against the lytic dsDNA *Pseudomonas* myoviruses PAP8 (*Pbunavirus*) and PaoP5 (*Pakpunavirus*) (Fig. 5a). At the single-cell scale, PaoP5 infection of WT PAO1 led to high levels of host cell lysis; in contrast, cell lysis was barely observed in cells expressing KKP$_{MP}$ (Fig. 5b). Expression of the K$^{GFI}$KP$_{MP}$ module conferred minimal protection from lysis, suggesting that kinase toxicity is important for phage defense. In timelapse microscopy studies, KKP$_{MP}$-bearing cells exhibited near-complete division arrest during lytic phage attack, whereas cell division was still observed in cells without KKP$_{MP}$ (Fig. 5c, d), suggesting that KKP$_{MP}$ protects cells from lysis by restraining cell growth and division.

To explore KKP$_{MP}$ antiphage activity in the native MPAO1 context, we screened a collection of diverse *P. aeruginosa*-targeting lytic phages against MP-ΔPf4ΔPf6 (lacks KKP$_{MP}$) and MP-ΔPf4ΔPf6core (retains KKP$_{MP}$) (Supplementary Fig. 8a). We identified two additional lytic dsDNA myoviruses, PAcq-4 (*Pbunavirus*) and PAgz-2 (*Phikmvvirus*), that appeared to be KKP$_{MP}$-sensitive and gave rise to visibly smaller plaques on MP-ΔPf4ΔPf6core lawns that contained a significantly reduced virion titer per plaque (Fig. 5e, f, Supplementary Fig. 8b). We observed virtually identical results in another MPAO1 derivative only lacking KKP$_{MP}$ (MPAO1 ΔKKP$_{MP}$). In static biofilm growth conditions, MPAO1 ΔKKP$_{MP}$ did not grow as well as MPAO1 when exposed to PAcq-4 phages, indicating that the lytic phage defense provided by KKP$_{MP}$ promotes biofilm integrity (Fig. 5g).

To examine the conservation of KKP phage defense activity outside of *P. aeruginosa*, we expressed KKP modules from four unrelated and non-filamentous prophages of *Shewanella* sp. W3-18-1, *Vibrio tasmaniensis*, *E. coli* 15EC039 and *Salmonella enterica* subsp. *enterica* serovar Typhi str. CT18 in a heterologous *E. coli* K-12 system. All 4 KKP modules conferred protection against phage T4 as well as at least one T-odd phage (T1/T5/T7), while they had no effect on EEP, λ, or the *E. coli* filamentous phage M13 (Fig. 6a). Further examination of KKP$_{SW}$ from *Shewanella* sp. W3-18-1, which is part of a P4-like prophage, confirmed that it is a tripartite TA system like KKP$_{MP}$ (Supplementary Fig. 9a, b). KKP$_{SW}$-expressing *E. coli* exhibited a marked suppression of phage production during infection with T1 or T4 (Fig. 6a, Supplementary Fig. 9c–e). This phenotype was dependent on the GFA motif in the PfkA$_{SW}$ CSD (analogous to the GFI motif in PfkA$_{MP}$) (Supplementary Fig. 9c–e). Similar to KK$_{MP}$ (Fig. 4c), phosphoproteomics of T4-infected, KKP$_{EC039}$-expressing *E. coli* revealed phosphorylation of many host factors important for cell growth and survival (Supplementary Fig. 9f). Notably, several host targets known to be important for phage propagation, including TopA[33,34], InfB[35] and RpoD[33,36] were also targeted by KK$_{MP}$ (Supplementary Fig. 9f). Collectively, these data indicate that diverse KKP modules provide phage defense, likely through a conserved mechanism of enacting growth arrest through PfkA/B-mediated phosphorylation of host targets needed for cell growth and lytic phage production.

## A phage replication protein activates KKP-mediated defense

To investigate the mechanism by which lytic phages trigger KKP defense, we isolated KKP$_{EC039}$-resistant mutants of T4 phage, which was highly susceptible to KKP-mediated defense (Fig. 6a) and has been extensively characterized[33]. Population and single-phage sequencing of evolved resistant phages revealed mutations in several T4 proteins (Fig. 6b–d, Supplementary Fig. 10a, b). Of these, only Gp59, the phage replication-related helicase loader[37] was able to specifically trigger KKP$_{EC039}$-mediated toxicity (Fig. 6e); moreover, the Gp59$^{G32E}$ mutation identified from the initial selection did not stimulate KKP$_{EC039}$ toxicity (Fig. 6e). These findings suggest that Gp59 activates KKP$_{EC039}$ defense during T4 infection. To examine if this phenotype could be due to direct binding of KKP components by Gp59, we performed pulldown assays and found that Gp59, but not Gp59$^{G32E}$, co-eluted with PfpC$_{EC039}$ (Supplementary Fig. 10c, Supplementary Table 11). If Gp59 binding to PfpC impaired its phosphatase activity, PfkA/PfkB kinase activity would increase and result in downstream cellular toxicity. Indeed, co-incubation of Gp59 with PfpC$_{EC039}$ modestly but statistically significantly lowered its phosphatase activity (Fig. 6f, Supplementary Fig. 10d). Consistent with this finding, structural modeling predicted that Gp59 can bind and potentially occlude the active pocket of the PfpC$_{EC039}$ phosphatase domain (Fig. 6g, Supplementary Fig. 10e). The Gp59$^{G32E}$ mutation is predicted to alter surface electrostatic potentials near the interaction surface, potentially explaining why Gp59$^{G32E}$ did not activate KKP$_{EC039}$ (Supplementary Fig. 10f). Since Gp59 is conserved in diverse phages[37] (Supplementary Fig. 10g), KKP may confer resistance against a range of lytic phages across a variety of hosts through a similar mechanism.

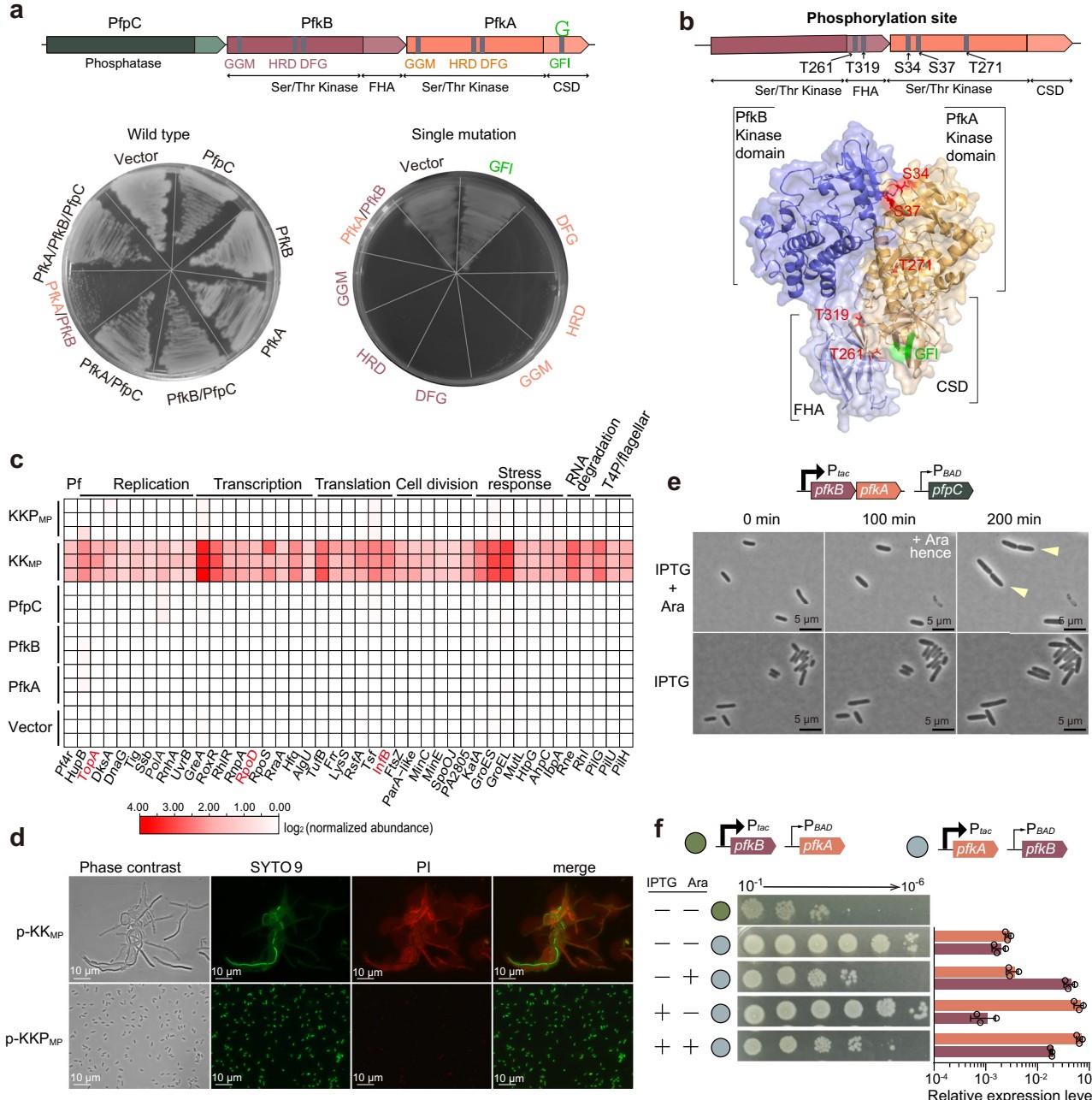

**Fig. 4 | KKP is a phosphorylation-based toxin/antitoxin system. a** Domain organization (top) of KKP$_{MP}$ and toxicity of PAO1 with expression of indicated KKP$_{MP}$ components with or without inducer (bottom). **b** AlphaFold2 prediction of the heterodimer structure of PfkA-PfkB complex. The identified phosphorylation sites in PfkA and PfkB are labeled in red (Supplementary Fig. 6d). **c** Heat map showing a subset of proteins with highest fold changes between KK$_{MP}$ *versus* KKP$_{MP}$ in PAO1 identified by phosphoproteomics (significant hits listed in Supplementary Table 5). **d** Live-dead staining of planktonic PAO1 cells overexpressing KK$_{MP}$ or KKP$_{MP}$. Cells were induced with IPTG for 6 h before staining. PI, propidium iodide. **e** Timelapse microscopy of immobilized *E. coli* MG1655 cells carrying KKP$_{MP}$ on a two-plasmid expression system, pTac-*pfkB*-*pfkA* (IPTG-inducible) and pHERD20T-*pfpC* (arabinose-inducible). Yellow arrows indicate dividing cells. **f** Dose-dependence of KKP$_{MP}$ toxicity in *E. coli* MG1655. *pfkA* and *pfkB* were cloned into two-plasmid systems (green and gray circles) with pTac (IPTG-inducible) and pHERD20T (arabinose-inducible) promoters. pTac has a higher leakage expression than that of pHERD20T. The top row without induction showed leaky kinase toxicity. Expression levels were evaluated by qRT-PCR and normalized to 16S rRNA gene expression levels. ND, not detected. Three independent cultures were used in (**a**, **c**, **d**, **f**), and representative images were shown in (**a**, **d**–**f**), and data are shown as the mean ± SD in (**f**).

## Discussion

Provision of both prophage regulation and phage defense by the tripartite KKP cassette is a novel union of two functionally disparate activities. The broad conservation of KKP modules in Gram-negatives suggests that this phosphorylation-based control system is a common mechanism of host-phage and host-prophage cross-talk. Nearly 90% of the >1200 identified KKP modules were in prophages, suggesting that this dual-functional cassette can be mobilized across diverse phyla. In addition, this regulatory module is often present in microbes that have co-resident prophages. Based on findings with the co-resident Pf6 and Pf4 in *P. aeruginosa*, we propose that KKP modules also mediate interactions between prophages in other organisms. However, given the identification of >100 KKP cassettes outside of prophages, often within other mobile elements, it is possible some KKP clusters control inter-MGE interactions, or only mediate phage defense. The functions of KKP in any

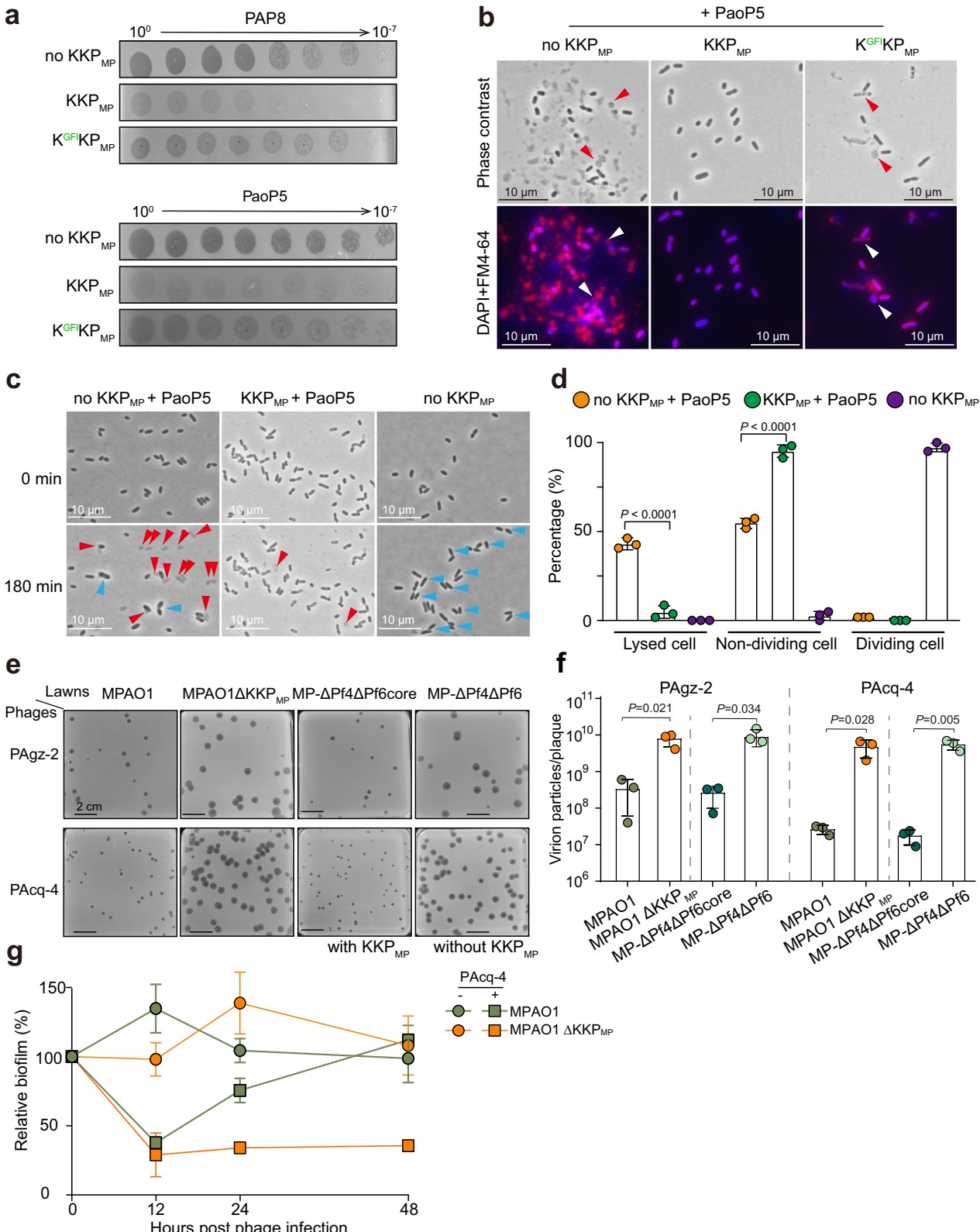

given bacterium are likely shaped by genomic context (i.e., prophage content), the frequency and type of lytic phage attack, as well as host-intrinsic phage control mechanisms.

Accessory genes in prophages, e.g., prophage-borne toxins like the cholera toxin genes in CTXΦ[2], often bestow critical properties to their hosts, but usually do not directly modify prophage development as we found with the Pf6 KKP module. We identified multiple potential host substrates for the PfkA/B kinases, among them the known Pf4 prophage silencing factor MvaU, and based on our data, propose the following mechanism for KKP control of Pf prophage induction. In planktonic cells, phosphorylation of MvaU by PfkA/B, which are highly expressed relative to *pfpC*, enforces repression of Pf genes required for prophage activation, thus maintaining Pf lysogeny. Conversely, in biofilms, expression of the entire KKP cassette, including *pfpC*, alters the balance

**Fig. 5 | KKP provides defense against phage infection through kinase toxicity.**
**a** Sensitivity of PAO1::$pilC_{MP}$ with chromosomally integrated $KKP_{MP}$ to different *P. aeruginosa* phages. PAP8 and PaoP5 are lytic phages. Expression of the phage receptor $pilC_{MP}$, which is truncated in PAO1, is required for filamentous phage infection and production in PAO1 cells. **b** Microscopy of PAO1 strains (the same as **a**) after 3 h infection with PaoP5 at MOI of 100. Red, FM4-64; blue, DAPI; arrows indicate lysed cells. **c** Time-lapse microscopy of immobilized cells (the same as **a**) infected with PaoP5. Red arrows indicate lysed cells and blue arrows indicate dividing cells. **d** Distribution of cells in different states (200 cells were counted) after infecting with PaoP5 for 180 min. Data are shown as the mean ± SD. Two-sided Student's *t* Test was used for comparisons between PaoP5 infection in the presence of $KKP_{MP}$ or not for each cell type ($n = 3$). $P < 0.05$ was considered statistically

significant. **e** Deletion of $KKP_{MP}$ in WT MPAO1 or MP-ΔPf4ΔPf6core increased plaque sizes of lytic *P. aeruginosa* phages PAgz-2 and PAcq-4. **f** Phages on plates in (**e**) were collected and the virion particles of each plaque were calculated. Data are shown as the mean ± SD. Two-sided Student's *t* Test was used for comparisons between two indicated groups in the presence of PAgz-2 or PAcq-4 lytic phages ($n = 3$). $P < 0.05$ was considered statistically significant. **g** Statically growing biofilms were treated with PAcq-4 phages, and the remaining biofilm was assessed at indicated time points to determine the relative biofilm compared to the initial time point (time 0) when the phages were added. Data are shown as the mean ± SD ($n = 3$). Three independent cultures were used in (**a**–**c**, **e**) and representative images were shown.

of kinase and phosphatase activities and alleviates Pf gene repression, presumably via dephosphorylation of MvaU. This leads to the production of Pf4 and Pf6 virions, which in turn alter the properties of *P. aeruginosa* biofilms, elevating antibiotic resistance and virulence (Fig. 6h)[4,10,12]. The concomitant induction of *pfpC* blocks kinase-based toxicity to enable temperate phage propagation during biofilm formation without negative effects on host cells. Prophage induction is often regulated through control of phage repressor activity, e.g., repressor cleavage by host factors[38]. The identification of KKP introduces a model for host-prophage interactions, since the 'lytic switch' is controlled by phosphorylation, a potentially reversible post-translational modification. This mechanism potentially enables dynamic prophage-host crosstalk and places KKP at the intersection of environmental signals, prophage activation, and host physiological state.

KKP is a phosphorylation-based tripartite TA system, where the PfkA and PfkB kinases act jointly as the toxin and the PfpC phosphatase as the antitoxin. Toxin neutralization via dephosphorylation suggests $KKP_{MP}$ is a type VII TA system, in which the antitoxin (PfpC) inactivates the toxin (PfkA/B) by post-translational chemical modification[39,40]. Activation of KKP in the context of T4 phage infection appears to occur by direct binding of PfpC to the phage replication protein Gp59, inhibiting phosphatase function and triggering unfettered kinase activity. Although the in vitro inhibition of $PfpC_{EC039}$ by Gp59 was modest, since PfpC lies at the top of the KKP signaling cascade, it is plausible that even relatively minor effects on phosphatase activity could feasibly result in the substantial impacts on downstream phosphorylation of host substrates we observed. Additionally, the true level of PfpC inhibition may be higher in vivo, as our data do not account for the possible contributions of additional host co-factors that may promote Gp59 binding to PfpC, or specific physiological biochemical conditions that PfpC may experience in the cell.

PfkA and PfkB target many essential cell processes to enact joint toxicity, including several required for phage propagation. As essential gene sets can vary between microbes, the precise mechanism of toxicity exerted by KKP may be species-dependent and further studies in orthogonal systems are needed to link kinase action to alteration of specific essential gene functions. KKP differs considerably from the only other known kinase-based phage defense system, Stk2 in *Staphyloccocus aureus*[41]. Chiefly, KKP is a prophage-encoded TA-based system that functions in prophage regulation as well as defense and is found in diverse Gram-negative species. In contrast, the S*taphylococcal* Stk2 system is host-encoded, consists of a single locus (the kinase), and appears to only contribute to phage defense. The identification of kinase-dependent defense systems in both Gram-positive and -negative species suggests Ser/Thr phosphorylation cascades may be a paradigm for bacterial responses to phage. Recent studies have identified both phage- and chromosome-encoded TA systems that mediate phage defense by abortive infection (Abi)[29,30]. Our findings suggest that KKP promotes defense through a similar process. However, unlike many other TA-based mechanisms of phage defense, KKP functions through post-translational modifications that can be controlled by the extent and duration of joint kinase activity by PfkA and

PfkB. In this sense, KKP may enable fine-tuned control of host cell fate during phage attack, similar to its tight control over prophage induction during biofilm development.

The discovery of KKP expands understanding of how mobile genetic elements such as phages can become entwined with microbial host pathways and modulate host phenotypes. In *P. aeruginosa*, KKP appears to provide common benefits for both prophage and host, which share the same chromosome, by restraining excessive prophage induction and protecting the cell from lytic phage attack. Identifying the pathways that relay distinct signals (e.g., phage attack or biofilm formation) to ultimately regulate KKP output may reveal triggers of phage defense and prophage induction. In particular, the precise environmental factors that trigger differential expression of KKP in planktonic and biofilm cells, which may include factors such as quorum sensing molecules and c-di-GMP, remain to be investigated. Given that several other newly identified phage defense systems are prophage-encoded[29,31,42–45], it will be of interest to investigate whether these systems also control additional types of phage-host or phage-phage interactions, such as prophage activation like KKP, or the establishment of lysogeny.

## Methods
### Bacterial strains, plasmids and growth conditions
The bacterial strains, plasmids and all primers used in the study are listed in Supplementary Tables 12, 13. *P. aeruginosa* PAO1 and MPAO1 sublines and *Escherichia coli* were cultured at 37 °C shaking at 220 rpm in Luria-Bertani (LB) medium. Gentamicin (Gm, 30 μg ml⁻¹) was used to maintain pEX18Gm-based plasmids, ampicillin (50 μg ml⁻¹) was used to maintain pEX18Ap, pFLP2 or pHERD20T-based plasmids, chloramphenicol (30 μg ml⁻¹) was used to maintain the pTac-based plasmids, and kanamycin (50 μg ml⁻¹) was used to maintain the pET28b-based plasmids. IPTG (0.1–1 mM) and *L*-arabinose (0.3%) were included to induce gene expression when needed.

### Flow cell biofilm assay
Flow cell biofilm assays were performed as previously described with minor modifications[19,46]. The system was assembled using sterilized material as shown in Fig. 1a. The assay was performed at room temperature with modified M9 medium containing 47.8 mM $Na_2HPO_4$, 22 mM $KH_2PO_4$, 6.8 mM $NH_4Cl$, 18.7 mM NaCl, 100 μM $CaCl_2$, 2 mM $MgSO_4$ and 0.1% glucose. For inoculation, 1 ml fresh overnight cultures were injected into the inlet of medical silicone catheters (3 mm inner diameter × 400 mm length) (Forbest Manufacturing Corporation, Shenzhen, China), taking care to avoid any bubbles. Inoculated catheters were left static for 1 h before flow initiation to allow colonization on the inner catheter surface. Then, flow was initiated with a peristaltic pump with a flow rate of 0.1 ml min⁻¹ for each channel. At indicated time points, phages were collected from the effluents and biofilm cells inside the catheter were collected by cutting the catheter open. Fresh M9 medium was supplied daily and total assay runtime lasted for 6–8 days depending on the readout.

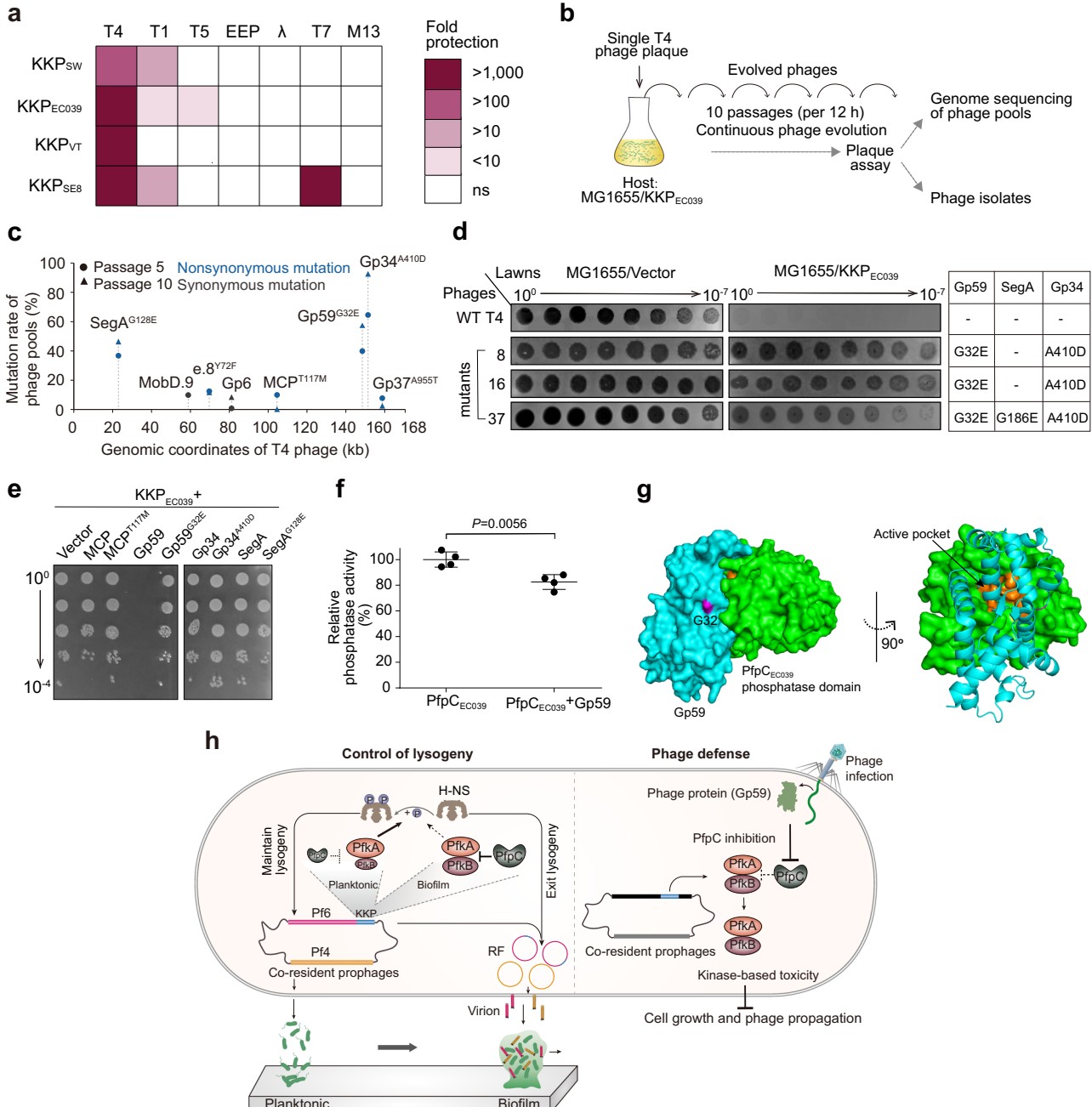

**Fig. 6 | Activation of a KKP cassette by a phage replication protein. a** Protection against phage infection by different KKP modules in *E. coli*. KKP$_{SW}$ *(Shewanella* sp.), KKP$_{EC039}$ *(E. coli* 15EC039), KKP$_{VT}$ *(Vibrio tasmaniensis)* and KKP$_{SE8}$ *(Salmonella enterica* CT8) are shown in blue in Fig. 3g. **b** Scheme for selecting T4 phage escape mutants. **c**, Genome sequencing of evolved T4 phage pools resistant to KKP$_{EC039}$ defense. Mutations with >20% occurrence were selected for follow-up. **d** Evasion of KKP$_{EC039}$-mediated defense by single isolated phages from (**b**). The table on the right shows the specific combination of mutations present in each phage. **e** Toxicity of co-expressing WT or mutant phage proteins and KKP$_{EC039}$ in *E. coli*. **f** Phosphatase activity of PfpC$_{EC039}$ in the presence or absence of Gp59. Data are shown as the mean ± SD (*n* = 4). Two-sided Student's *t* Test was used for statistical analysis (*n* = 4). *P* < 0.05 was considered statistically significant. **g** Structural model

of the PfpC$_{EC039}$-Gp59 complex. G32 in Gp59 is displayed in purple. Gp59 (in light blue) interacts with the PfpC phosphatase domain (in green) and occludes its active pocket (in orange). The confidence level of the predicted complex has a predicted local distance difference test (pLDDT) score of 82. **h** Model for dual control of lysogeny and phage defense by KKP. KKP controls the two sister prophages Pf4 and Pf6 through regulated phosphorylation of a host H-NS protein. KKP also provides defense against lytic phage through kinase-based toxicity triggered by a phage protein, leading to inhibition of cell growth and/or phage propagation. Arrows and "T" symbol indicate stimulation and inhibition, respectively. Dashed lines indicate that influence is weakened or eliminated. RF, replicative form. Three independent cultures were used in (**a**, **e**).

## Pf phage quantification by plaque assay

Phage plaque assays were conducted using the top-layer agar method[47] to quantify the number of Pf phages. Briefly, 2 ml of biofilm effluents were collected at day 6 post-inoculation and centrifuged at 13,000 × *g* for 5 min. The obtained supernatants were filter sterilized with 0.22 μm filters (Millipore Corporation, Villerica, MA, USA) to obtain cell-free phage solutions. Since high titer Pf phages are known to form clustered liquid crystals that could impact phage quantification[10], phage samples were serially diluted in M9 buffer containing 1% methanol combined with vortexing to prevent

liquid crystal formation. 5 μl of 10-fold dilutions were spotted on a R-top layer by mixing fresh overnight cultures with medium containing 0.8% (w/v) agar, 0.1% yeast, 1% tryptone, and 1% NaCl. The R-bottom layer contained 1% (w/v) agar, 0.1% yeast, 1% tryptone and 0.8% NaCl. After drying, plates were cultured at 37 °C for 16 h and the numbers of plaques formed were counted. For plaque assays involving overexpression genes, overnight cells were diluted and cultured as above to $OD_{600}$ ~ 1. IPTG was then added for 2 h to induce gene expression, after which cells were mixed with R-top medium containing IPTG and plaque assays were performed as above.

## Pf phage quantification by quantitative PCR (qPCR)

We optimized previously described qPCR assay[48] to obtain a more accurate estimation of Pf4 and Pf6 abundance in biofilm effluents, using primers amplifying the Pf4-specific gene *pfiA* and Pf6-specific gene *intF6* listed in Supplementary Table 13. First, the amplification efficiencies of the designed primers were assessed through serial dilutions of linearized pMD19-*pfiA* and pMD19-*intF6* plasmids (Supplementary Fig. 3c). Next, Pf4 and Pf6 phages collected from effluents of MP-ΔPf4 and MP-ΔPf6core biofilms were propagated by infecting exponential-growing MP-ΔPf4ΔPf6 cells ($OD_{600}$ ~ 1.0) for 6 h (Supplementary Fig. 3d). High titers of Pf4 and Pf6 in the supernatants were then obtained by centrifugation and filtration through a 0.22 μm filter (Millipore). The filtered phages were treated with 1 μg/ml DNase I and 1 μg/ml RNase I (New England Biolabs, MA, USA) at room temperature for 2 h[49]. Lastly, the Ct values of *pfiA* and *intF6* were determined using the serial phage dilutions as templates, and we found that Ct values are linearly correlated to Pf phage titers ranging from $10^5$ PFU/ml to $10^{12}$ PFU/ml with $R^2 > 0.98$ (Supplementary Fig. 3e).

To quantify Pf gene copy numbers in host cells, host genomic DNA was isolated using the TIANamp Bacteria DNA kit (Tiangen Biotech Co. Ltd, Beijing, China) according to the manufacturer's instructions, and 20 ng gDNA was used as input for qPCR. When necessary, the levels of Pf4 *attP* were also determined. *gyrB* was used as a housekeeping gene, and amplification efficiencies of all primer pairs used were checked ($R^2 > 0.98$).

## Microscopy

For transmission electron microscopy (TEM), phages in filtered effluents at day 6 were negative stained with 2% phosphotungstic acid. Phage ultrastructure was then imaged with a TEM H-7650 (Hitachi) instrument at the Guangdong Institute of Microbiology (Guangzhou, China). For fluorescence microscopy of GFP-tagged MCP (pVIII), biofilms formed in medical catheter at day 6 were collected and observed with a Scope A1 fluorescence microscope (Carl Zeiss, Jena, Germany). For fluorescence microscopy during anti-phage assays, overnight cells were diluted and grown at 37 °C till $OD_{600}$ ~ 1. Phages were then added to cultures at the indicated MOIs for 3 h. Cells were then stained with FM4-64 (0.5 μg ml⁻¹) and DAPI (1 μg ml⁻¹) for 10 min and subsequently imaged. For HADA staining, overnight cultures were diluted to $OD_{600}$ ~ 0.1, then *L*-arabinose and 200 μM HADA were added for 4 h prior to observation with fluorescence microscopy. Time-lapse microscopy was performed using a 1.5% agarose pad prepared with LB containing the inducer.

## Dose-dependent toxicity assays

Dose-dependent toxicity of KKP$_{MP}$ was performed using two-plasmid system with pTac and pHERD20T promoters. For measurements of CFU, gene expression was induced using IPTG or *L*-arabinose for 4 h. For mRNA analysis, gene expression was induced using IPTG or *L*-arabinose for 0.5 h.

## Chromosomal gene knockout and insertion in *P. aeruginosa*

A previously described gene knockout method with modifications was used to construct chromosomal gene deletion and insertions in *P. aeruginosa*[50]. All primers used are listed in Supplementary Table 13. For the deletion strains, the upstream and downstream fragments of each gene were PCR-amplified from MPAO1/PAO1 genomic DNA. Gel-purified amplicons were then ligated into the modified suicide plasmid pEX18Gm. The constructs were confirmed by sequencing using primers pEX18Gm-F/R. Then, deletion vectors were transferred into *E. coli* WM3064, a diaminopimelic acid (DAP) auxotroph, and conjugated into MPAO1/PAO1 strains. In-frame deletion mutants were obtained via homologous recombination using the sucrose resistance selection method. For strains of PAO1::*pilC*$_{MP}$::KKP$_{MP}$, and PAO1::*pilC*$_{MP}$::K$^{GFI}$KP$_{MP}$, the expression of KKP$_{MP}$ or K$^{GFI}$KP$_{MP}$ used the promoter from MPAO1. The coding region of KKP$_{MP}$ and the upstream 400 bp was fused with the Gm resistance gene and cloned into pEX18Ap. For MPAO1::*mvaU*-His strain, the coding region of *mvaU* and C-terminal 6 x His was fused with the Gm resistance gene. The Gm resistance cassette was removed from the chromosome using pFLP2 as described previously with minor modifications[50]. In brief, the pFLP2 was transferred into Gm resistance strain to remove Gm markers by recombinase-catalyzed excision, and the Gm losing single colonies were confirmed by screening on LB plates with and without Gm antibiotics. The pFLP2 plasmid in the final correct Gm sensitive colonies was cured by LB plates supplemented with 5% sucrose. The loss of pFLP2 was further determined on LB plates containing Ampicillin or Gm. Final deletion mutants were confirmed by PCR using primers gene-SF/SR, gene-LF/LR and DNA sequencing.

## Antibiotic resistance assay

To assess the antibiotic resistance of planktonic cells, exponentially growing cells ($OD_{600}$ ~ 1) were treated with 20 μg ml⁻¹ gentamicin (Gm, 40 x MIC) for 1 h. Colony formation units (CFU) were then determined by serial plating. To assess antibiotic resistance in biofilm-state cells, attached biofilms were collected by carefully cutting 1 cm length of medical silicone catheters at day 6 post inoculation. The catheters were then divided into two groups: intact biofilm and destroyed biofilm. For the intact biofilm group, catheters were immersed in 3 ml LB containing Gm in a shaking incubator with 220 rpm for 3 h at 37 °C. For the destroyed biofilm group, biofilm cells inside the catheters were collected by scraping into 3 ml LB using pipette tips, followed by mechanically disruption achieved through by repeated pipette aspiration and vigorous vortexing. This disruption process facilitated the release of the majority of biofilm cells, as confirmed by microscopy. The mechanically disrupted biofilm cells were then exposed to Gm in a shaking incubator at 37 °C with 220 rpm for 3 h.

## In vivo infections

All animal experiments were reviewed and approved by the Animal Ethical and Welfare Committee of Blood Disease Hospital, the Chinese Academy of Medical Sciences (IHCAMS-DWLL-TJ2022012-1). Eight-week-old WT C57BL/6J female mice were used in all experiments and were purchased from the Jackson Laboratory. Only female mice were used as we did not investigate the influence of the mice sex difference on *P. aeruginosa* in this study. Mice were housed in pathogen-free cages under 12/12 h dark/light cycle with adequate food and water and comfortable bedding. The ambient temperature was 20–22 °C. The humidity of the housing conditions was controlled at about 60%. All necessary measures were taken to avoid or minimize any discomfort, pain, or distress to the animals. During pathogen instillation surgery, mice were anesthetized by intraperitoneal injection of 2,2,2-tribromoethanol (250 mg kg⁻¹ body weight). The ventral cervical region was surgically prepared, a 0.5-cm skin incision was made at the thoracic inlet, and the trachea was visualized by blunt dissection. Mice were inoculated with an angiocatheter directed into the left main-stem bronchus. A 24 G

0.7 mm × 19 mm I.V. angiocatheter was used to cannulate the trachea. A one milliliter syringe was pre-loaded with 0.1 ml of air to account for dead space, and then filled with 50 μL of bacteria (planktonic or biofilm embedded in agarose beads, see below) suspension for intratracheal inoculation. Mice for the planktonic model were challenged with $2-3 \times 10^6$ planktonic bacteria. Mice for the biofilm model were challenged with $1-4 \times 10^6$ bacteria embedded in agarose beads. Incisions were closed with suture clips and mice were placed on a heating pad until awake. All mice were monitored for 120 h for survival.

For the biofilm inocula, *P. aeruginosa*-laden agarose beads were prepared one day prior to infection as previously described[51]. Briefly, bacteria were grown to late log phase in a shaking incubator at 37 °C. 20 ml of low EEO agarose in phosphate buffered saline (PBS), pH 7.4 was mixed with a 4 ml aliquot of the culture for a final agarose concentration of 1.2%. The mixture was added to heavy mineral oil that was equilibrated to 55 ± 0.5 °C, rapidly stirred with a magnetic stirring bar for 6 min at room temperature and then cooled for 20 min. The agarose beads were washed once with 0.5% sodium deoxycholate (SDC) in PBS, once with 0.25% SDC in PBS, and 4 times with PBS alone. The bead slurry was allowed to settle so that 75% of the final volume consisted of beads. CFU plating was performed on an aliquot of homogenized bead slurry to determine inoculum size. Bead diameter was measured using an inverted light microscope in several fields, and the diameters of beads was between 100 and 200 μm.

### Construction of plasmids and site-directed mutagenesis

The construction of all pHERD20T-based plasmid digested with NcoI and HindIII has been described previously[13]. Specifically, the coding regions of candidate genes were PCR amplified with primers listed in Supplementary Table 13, and the gel purified amplicons were ligated into the NcoI and HindIII digested pHERD20T vector. The correct clones were confirmed by both PCR and DNA sequencing. For the cloning of genes with specific site mutagenesis, previously reported procedures were used[52]. Similar procedures were used for cloning indicated genes into pTac plasmid digested with EcoRI and BamHI. The KKP_{VT} and KKP_{SE8} amplicons for cloning were synthesized by Tsingke Biotechnology (Beijing China) Co., Ltd. and inserted into the same sites of pTac.

### de novo sequencing and assembling of PAO1 and MPAO1 genomes

For genome sequencing of PAO1 and MPAO1, genomic DNA were isolated from exponential phase (OD_{600} ~ 1) cells. gDNA was further purified using Agencourt AMPure XP Kit (Beckman Coulter). The purity, quantity, and size of DNA was assessed through a combination of NanoDrop, Qubit and pulsed field gel electrophoresis. Following quality control, 5 μg DNA was used to prepare 20 kb SMRTbell libraries per the manufacturer's directions (Tianjin Biochip Corporation, Tianjin, China). Subsequently, the quantity and size of each library was assessed and sequenced on a Pacific Biosciences (PacBio) RSII instrument using P6-C4 chemistry with 6 h movie time following the manufacturer's recommendations. A total of 108,077 reads for PAO1 and 32,240 reads for MPAO1 were obtained, and they were assembled using HGAP algorithm version 4 into one contig with average genome coverage of 800 × and 300 × respectively. The genome sequences of MPAO1 and PAO1 were compared with the first sequenced PAO1 genome (GenBank ID: AE004091) and MPAO1 (GenBank ID: PRJNA438597). The genome sequences of MPAO1 in this study was used as a reference. Images showing the four genome comparisons were generated using BLAST Ring Image Generator (BRIG) (version 0.95-dev.0004)[53].

### Quantitative real-time reverse-transcription PCR (qRT-PCR)

Total RNA in planktonic and biofilm cells of PAO1 and MPAO1 were isolated using the Bacteria RNAprep pure kit (Tiangen Biotech Co.

Ltd, Beijing, China) according to the manufacturer's instructions. The MPAO1 ΔmvaUΔmvaT strain with pHERD20T-based constructs to produce MvaU mutants were induced with *L*-arabinose for 3 h and total RNAs were also isolated as above. A total of 200 ng total RNA was used for reverse transcription reactions to synthesize cDNA with reverse transcription system A3500 (Promega, Madison, WI). Then, 50 ng of cDNA was used to conduct qRT-PCR with SYBR green reaction mix on a Step One Real-Time PCR System (Applied Biosystems). 16S rRNA gene levels were used to normalize expression levels of genes.

### Strand-specific RNA sequencing

Overnight PAO1 and MPAO1 planktonic cultures in M9 medium (OD_{600} ~ 3) were collected by centrifugation at 12,000 *g* for 30 s at 4 °C. The biofilm cells on the inner surface of the medical catheter were collected at day 6 by cutting 5 mm at the middle of the catheter, and cells were collected by centrifugation. Total RNA was isolated and quality control was performed with Agilent RNA ScreenTape (Lot: 0201937-152). rRNA was removed with the Ribo-off rRNA Depletion Kit (Bacteria) (Vazyme, # N407-C1). To prepare cDNA libraries, 1 μg total RNA was used for cDNA synthesis amplification (11 cycles) using VAHTS Total RNA-seq (Bacteria) Library Prep Kit for Illumina (Lot:7E582A1) according to the manufacturer's instructions. Then, the libraries were sequenced on a NovaSeq 6000 S4 (Illumina). The obtained raw data were evaluated with FastQC (v0.10.1), and contamination and adapter sequences were removed with Cutadapt (version 1.9.1). Cleaned data were mapped to the MPAO1/PAO1 genome with bowtie2 (v2.2.6), and the expression profiling was calculated with Htseq (V 0.6.1). The differential gene expression analysis was performed using EdgeR[54] (R-3.1.2), and significantly changed genes (≥2-fold and FDR ≤ 0.05) in biofilm compared to planktonic cells were selected for further analysis.

### Phosphoproteomic analysis

Different KKP_{MP} components were expressing using pHERD20T-based plasmids in MPAO1 or PAO1 with *L*-arabinose. Cell pellets were collected by centrifugation at 6000 *g* for 5 min at 4 °C and washed twice. KKP_{EC039} was expressed via pTac-based plasmid with 0.4 mM IPTG. For the cells with T4 infection, phages were added simultaneously with IPTG and the MOI was ~10. Lysate preparation and LC-MS/MS analysis were carried out by Jingjie PTM BioLab Co. Ltd. (Hangzhou, China). Samples were sonicated 25 times at 220 W (3 s per round, 5 s interval) on ice in lysis buffer (8 M urea, 1% protease inhibitor cocktail, 1% phosphatase inhibitor). The remaining debris was removed by centrifugation at 12,000 *g* at 4 °C for 10 min. Protein concentration was determined with the bicinchoninic acid assay according to the manufacturer's instructions. The procedures of protein trypsinization, phosphopeptide enrichment, LC-MS/MS, and data analysis were conducted as previously described[55,56]. Briefly, obtained protein were digested in trypsin solution for 12 h with 50:1 (m/m) protein to protease ratio. Then 5 mM dithiothreitol was added and incubated at 56 °C for 30 min followed by adding 11 mM iodoacetamide and incubating at room temperature in dark for 15 min. The digested peptides were dissolved in the enrichment buffer containing 50% acetonitrile. The modified peptides were extracted with 10% ammonia water vacuum-dried. After that, the obtained peptides were used for tandem MS/MS in Q Exactive Plus coupled online to the UPLC. The resulting MS/MS data were processed using MaxQuant search engine (v.1.6.15.0), and the tandem mass spectra were searched against the MPAO1 or *E. coli* MG1655 proteomes.

### Protein purification and LC-MS/MS analysis

C-terminal 6 x His-tagged MvaU was purified from MPAO1::*mvaU*-His cells expressing pHERD20T-*pfkA*, pHERD20T-*pfkB*, pHERD20T-*pfkA*-*pfkB* or pHERD20T-KKP. All the strains were cultured in LB with

ampicillin and induced with *L*-arabinose for 3 h. Cells were collected and His-tagged proteins were purified as described previously[52]. Briefly, the cell pellets were lysed in lysis buffer containing 50 mM potassium phosphate, 300 mM NaCl, and protease inhibitor cocktail (Sigma-Aldrich, United States) using a Sonic Dismembrator (Ningbo Dongzhi, China) at level 2 for 5 min on ice. Ni-NTA resin (Qiagen) was used to bind the His-tagged protein and proteins were eluted with 20 mM Tris-HCl buffer (pH 8.0) with increasing concentrations of imidazole. PfkA-His and His-PfkB were purified from *E. coli* Rosetta cells harboring pET28b-*pfkA*-His and pET28b-His-*pfkB*, respectively. Both strains were induced with 0.5 mM IPTG for 3 h and protein were purified with above method with 7 M urea in the buffers. Phosphorylation sites of the purified MvaU-His in different hosts, purified PfkA-His and His-PfkB were identified using LC-MS/MS as previously described[56] and above procedures for phosphoproteomic analysis. The tandem mass spectra were searched against the corresponding protein sequences.

## Bioinformatic analysis of KKP modules

KKP modules were searched using a triplet neighboring gene strategy. First, all genes with a kinase domain (COG0515 or PF00069, short name: Pkinase) or a phosphatase domain (COG0631 or PF13672, short name: PP2C) in 86,934 genomes in the IMG/M database (Supplementary Data 3) (including prokaryotic and eukaryotic genomes) were extracted using the Function Profile analysis tool[57]. All retrieved genes were then filtered by gene neighborhood and orientation, namely triplet neighboring genes with the order of PP2C, Pkinase and Pkinase. Hits were then further filtered by gene length (<1300 bp and >900 bp) to give the list of putative KKP clusters. The filtered sequences of PfkA-like proteins were aligned by MAFFT[58] and further edited by trimAl[59]. The final data set was used for maximum likelihood (ML) phylogenetic analysis by W-IQ-TREE[60]. The best-fit substitution model was automatically determined and the reliability of internal branches was tested by 1000 ultrafast bootstrap replicates[61] in the W-IQ-TREE web interface. Eukaryotic kinases (Uniprot IDs: Q9SMQ4, Q9ES74 and Q8TDX7) were used as the outgroup. The tree was further annotated by iTOL[62]. Genomic locations of KKP gene clusters were next evaluated with Shell commands to determine whether they were in a predicted prophage region or adjacent to MGE-related genes (upstream- or downstream- 5 genes or 3000 bp). Prophages were predicated by Virsorter[63] and MGE-related genes were predicated using mobileOG-db database[64]. We found that Virsorter frequently missed identification of P4 prophages. Therefore, P4 prophages were specifically identified by searching with the query sequence for P4 Psu (phage polarity suppression protein, PF07455). Similarly, known integrases, RM systems, TA systems and transposases that were missed by mobileOG-db searches were further identified by manually checking the gene annotation from the IMG//M database.

## Protein 3D structure prediction

Protein structures of PfkA/PfkB/PfpC and PfkA-PfkB complex were predicted with AlphaFold2 (v2.0.1)[65], and Dali server[66] was used to search for structural homology. For structural alignment, we used PknB from *Mycobacterium tuberculosis* (PDB ID: 1O6Y) as a template for PfkA and PfkB, and PP2C from *Thermosynechococcus elongatus* (PDB ID: 2J86) as a template for PfpC. The interaction between Gp59 and phosphatase domain of PfpC$_{EC039}$ was also predicted with AlphaFold2. Structural figures were produced with PyMOL (www.pymol.org).

## Biofilm degradation assay

The experiment was conducted as described previously[67]. 200 μl aliquots of MPAO1 or MP-ΔKKP$_{MP}$ (OD$_{600}$ ~ 0.1) were incubated in 96-well microtiter plates for 24 h to form statically growing biofilms. Then, the planktonic cells were removed carefully and 200 μl of PAcq-4 diluted

in LB broth was added to each well at MOI 0.1. The biofilm was determined at 12, 24 and 48 h post phage infection by crystal violet staining.

## Growth curves after phage infection

Overnight *E. coli* cultures harboring the indicated KKP and mutant plasmids were diluted to OD$_{600}$ ~ 0.2. 100 μl of culture was seeded into 96-well plates, then 100 μl of indicated phages was mixed with seeded cells in LB medium containing ampicillin and *L*-arabinose to obtain MOI 0.1. The mixtures were allowed to grow at 37 °C without shaking, and OD$_{600}$ was measured at the indicated timepoints.

## One-step growth curves for measuring anti-phage activity

The experiment was conducted as described recently[29]. Specifically, cells with the indicated plasmids were infected with phages at the indicated MOI in LB medium containing ampicillin and *L*-arabinose. Phages were collected at indicated time points by centrifugation at 2400 *g* for 5 min and sterile-filtering the supernatant with 0.22 μm filters. The obtained phages were serially diluted and immediately dropped onto top medium containing 0.5% agar and *E. coli* MG1655 cells to quantify PFUs.

## Evolution of mutant phages that overcome defense

To experimentally evolve phages to escape KKP defense, KKP$_{EC039}$, which conferred the highest protection against T4 lytic phage, was chosen. The general procedure was modified from a previous study[31]. For each passage, overnight *E. coli* MG1655 cultures expressing KKP$_{EC039}$ were diluted 100-fold and induced with 0.3% *L*-arabinose, T4 phages were added at OD$_{600}$ ~ 1.0 at MOI 0.1, and then cultured at 37 °C at 220 rpm for 12 h. Phages were collected by centrifugation of cultures at 10,000 *g* for 1 min, and collected supernatants were filtered through 0.22 μm filters. The PFU in the supernatants were determined using the top-layer plaque assay. Supernatants with evolved T4 populations were used for evolution of the next passage for a total of ten passages.

## Sequencing and genome analysis of mutant phages

Genomic DNA from T4 phage pools at passages 0, 5 and 10 were isolated using the TIANamp Virus DNA Kit (Tiangen, Nanjing, China) according to the manufacturer's procedure. Libraries were prepared for Illumina paired-end sequencing (PE150), and sequenced by Shanghai Biozeron Biotechnology Co., Ltd (Shanghai, China). Raw paired end reads were trimmed and quality controlled by Trimmomatic with default parameters (http://www.usadellab.org/cms/uploads/supplementary/Trimmomatic). High quality sequencing reads were then aligned to the T4 phage reference genome (NCBI accession number NC_000866) using BWA with "bwa mem" mode. After removing PCR-duplication reads with SAMtools (http://samtools.sourceforge.net/) software, the resulting BAM file was used to detect SNPs and short indels with the GATK "HaplotypeCaller" function (http://www.broadinstitute.org/gatk/). Variant call format (VCF) files were generated and filtered with VCFtools (version 0.1.11). Annotation of detected variations was performed by ANNOVAR. Structure variations (SV) were identified by BreakDancer (http://breakdancer.sourceforge.net).

## Pulldown assay

N-terminal 6 x His-tagged PfpC$_{EC039}$ was cloned into pETDuet digested with NcoI and HindIII to construct pETDuet-His-PfpC$_{EC039}$, and it was then digested by NdeI and XhoI. Next, Gp59 or Gp59$^{G32E}$ was introduced to obtain pETDuet-His-*pfpC*$_{EC039}$-*gp59* or pETDuet-His-*pfpC*$_{EC039}$-*gp59*$^{G32E}$. For purification of Gp59 and Gp59$^{G32E}$, the sequences of the two genes were amplified with WT or escape T4 mutants as templates and ligated into pET28a. Pulldowns with the cloned genes were performed as described previously[68]. Briefly, His-PfpC$_{EC039}$ and

Gp59/ Gp59$^{G32E}$ proteins were co-expressed via pETDuet-His-*pfpC*$_{EC039}$-*gp59* or pETDuet-His-*pfpC*$_{EC039}$-*gp59*$^{G32E}$ with above conditions for protein purification, and the following processes for protein complex purification are also the same.

## Phosphatase activity assay

The phosphatase activity of PfpC$_{EC039}$ in the presence or absence of Gp59 was determined using the Beyotime Alkaline Phosphatase Assay Kit (Nanjing, China) following the manufactures' instructions. Purified PfpC$_{EC039}$ was premixed with Gp59 and incubated on ice for 30 min to allow the formation of the complex.

## Statistical analysis

For all experiments, at least three independent biological replicates were used unless otherwise noted. Significance testing was performed by an unpaired $t$ test for comparisons between two groups, and a one-way analysis of variance (ANOVA) test with Tukey's correction for multiple comparisons. The log-rank (Mantel-Cox) test was used to analyze survival percentages of mice after infection. A $P$ value of < 0.05 was considered as statistically significant.

## Reporting summary

Further information on research design is available in the Nature Portfolio Reporting Summary linked to this article.

## Data availability

The data that support the findings of this study are provided within the manuscript and its associated Supplementary Information. The whole bacterial genome sequences of MPAO1 and PAO1 have been deposited in NCBI's GenBank Database under BioProject PRJNA745538, and the accession numbers are CP079712 and CP085082. They were annotated by combination of the annotation to MPAO1 with GenBank accession number PRJNA438597 and RASTA rapid annotation using subsystem technology (version 2.0)[69]. In addition, the draft genome of *E. coli* 15EC039 has been submitted to the same database under accession number JAMFTL000000000. RNA-seq raw data were deposited in Sequence Read Archive (SRA) under BioProject accession number PRJNA835970. The mass spectrometry data of purified proteins and phosphoproteomic data have been deposited to the ProteomeXchange Consortium via the PRIDE partner repository[70]. The mass spectrometry data of MvaU purified from different hosts are under projects PXD035222 and PXD044529, the phosphoproteomics data for PfkA overproduction in MPAO1 is under project PXD035424, the phosphoproteomics data for KKP components in PAO1 are under projects PXD035228 and PXD043326, and the phosphoproteomics data for MG1655 with KKP$_{EC039}$ overexpression combined T4 infection is under project PXD053915. There are no restrictions on data availability. Source data are provided with this paper.

## Code availability

The script used for searching KKP with triplet gene strategy can be accessed at https://github.com/WangLab-SCSIO/KKP_search.

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

## Acknowledgements

This work was supported by the National Science Foundation of China (42188102 to X.W., 92451302 to Y.G., 31625001 to X.W., 91951203 to

X.W., 42376128 to K.T. and 31970037 to Y.G.), by the Science & Technology Fundamental Resources Investigation Program (2022FY100600 to X.W.), by the National Science Foundation of Guangdong Province (2024A1515011146 to Y.G.), by the Guangdong Major Project of Basic and Applied Basic Research (2019B030302004 to Y.G.), by the Guangdong Local Innovation Team Program (2019BT02Y262 to X.W. and Y.G.), and by the Tianjin Municipal Science and Technology Commission Grant (21JCQNJC01550 to F.L.). We thank Prof. Tao G. Dong for kindly sharing the PAO1 subline, Profs. Yingfei Ma and Shuai Le for kindly sharing the lytic *P. aeruginosa* phages used in this study. The Waldor lab is supported by the Howard Hughes Medical Institute (HHMI). This article is subject to HHMI's Open Access to Publications policy. HHMI lab heads have previously granted a nonexclusive CC BY 4.0 license to the public and a sublicensable license to HHMI in their research articles. Pursuant to those licenses, the author-accepted manuscript of this article can be made freely available under a CC BY 4.0 license immediately upon publication.

## Author contributions

X.W., M.W., and Y.G. conceived the project. Y.G., K.T., B.S., and J.G. contributed equally to this work. Y.G., J.G., J.L., S.L., Z.H., X.L., W.W., X.G., Z.N., and T.L. constructed strains, performed all other wet-lab experiments, K.T., Y.G., and R.C. conducted all bioinformatic analyses. H.L., X.S., and F.L. performed the mice infection experiments. All authors interpreted data, X.W., M.W., B.S., Y.G., and K.H.T. wrote the paper.

## Competing interests

The authors declare no competing interests.

## Additional information

[1]Key Laboratory of Tropical Marine Bio-resources and Ecology, Guangdong Key Laboratory of Marine Materia Medica, Innovation Academy of South China Sea Ecology and Environmental Engineering, South China Sea Institute of Oceanology, Chinese Academy of Sciences, Guangzhou, China. [2]Southern Marine Science and Engineering Guangdong Laboratory (Guangzhou), Guangzhou, China. [3]University of Chinese Academy of Sciences, Beijing, China. [4]Department of Microbiology, Harvard Medical School, Boston, MA, USA. [5]Division of Infectious Diseases, Brigham and Women's Hospital, Boston, MA, USA. [6]State Key Laboratory of Experimental Hematology, National Clinical Research Center for Blood Diseases, Institute of Hematology and Blood Diseases Hospital, CAMS Key Laboratory for Prevention and Control of Hematological Disease Treatment Related Infection, Chinese Academy of Medical Sciences and Peking Union Medical College, Tianjin, China. [7]Boston Children's Hospital, Dana-Farber/Harvard Cancer Center, Boston, MA, USA. [8]Howard Hughes Medical Institute, Bethesda, MD, USA. [9]Present address: Department of Biology, Massachusetts Institute of Technology, Cambridge, MA, USA. [10]These authors contributed equally: Yunxue Guo, Kaihao Tang, Brandon Sit, Jiayu Gu. ✉e-mail: mwaldor@bwh.harvard.edu; xxwang@scsio.ac.cn

