## [Peer Review File · Nature Communications]

Control of lysogeny and antiphage defense by a prophage-encoded kinase-phosphatase moduleEditorial Note: This manuscript has been previously reviewed at another journal that is not operating a transparent peer review scheme. This document only contains reviewer comments and rebuttal letters for versions considered at *Nature Communications*.

Reviewer #1 (Remarks to the Author):

The authors have responded to the initial issues raised and improved the manuscript. There are, however, remaining issues and concerns. The key concerns can perhaps best be summarized by highlighting two problematic passages of the text, one in the Abstract and one in the Discussion: In the abstract the authors write: "The balance between kinase and phosphatase expression regulates phage production by controlling the phosphorylation of a host nucleoid binding protein." As discussed below, it is not rigorously established that it is the balance in expression that is ultimately responsible for changing the phosphorylation of MvaU, let alone other kinase targets. The expression ratio absolutely does change but whether it is *responsible* is simply not shown. This sort of imprecision in logic and writing really holds the paper back. Additionally, as written, this sentence in the abstract also implies that MvaU is the only or the primary target, which is also not rigorously established. Indeed, the phosphoproteomics suggests multiple targets, most of which haven't been examined in depth. Then, in the Discussion, the authors prematurely conclude/summarize that: "Phosphorylation of a host H-NS protein (MvaU) by PfkA/B represses expression of Pf genes required for prophage activation, keeping Pf6 and Pf4 gene expression largely silent in planktonic cells. However, in biofilms, expression of the entire KKP cassette, including PfpC, alleviates Pf gene repression, presumably via dephosphorylation of MvaU." The last sentence, in particular, is not firmly established yet - the authors do say 'presumably' via dephosphorylation of MvaU, but this really should be established (or not, as it may turn out). And this last quoted sentence also brushes under the rug the issue regarding how kinase/phosphatase balance changes and whether it's a change in expression of kinase, phosphatase, or both, or whether there are other factors involved. My primary concerns and a few other minor ones are further described below:

line 38: "provides defense against lytic phages". Against some lytic phages. As written, it implies defense against all lytic phages. Again, this lack of precision in language was problematic in the first version of the paper and continues to plague this one.

The titer of the MPAO1 WT is similar to that of the pf4 and pf6 single deletions. But what phages are produced by the WT? It could be a mix of both, but there could also be one that dominates. It should be straightforward to assess this and report it. This is partially addressed in the responses to reviewers document, but I don't see a clear explanation or the relevant data in the revised paper.

* lines 127-139: This is a major issue. As written, this section concludes that the change in expression of the phosphatase relative to the kinases is *responsible* for the induction of prophages in biofilms. While the expression ratio does change and a null mutant of kinase A shows a clear phenotype, these data alone do not allow one to conclude that it is the change in RNA levels that is responsible. To make that conclusion, the authors need to show that a mutation or genetic change preventing only the change in expression ratio has the same phenotype as the null mutant.

* Related to the point above: it remains unclear what triggers the KKP-dependent induction of Pf4 in biofilms. In other words, what causes the change in expression ratio, if that's indeed what's responsible for the induction? Or, if that's not what ultimately drives induction, what is responsible? (It should also be noted that in Fig. 2b it's not just the ratio of phosphatase to

kinase that changes in biofilms, but the relative expression of the kinases. Moreover, it must be emphasized that these are RNA levels - we don't know what's happening at the protein level at all.)

The model in Fig. 2h posits that MvaU is phosphorylated by the A/B kinases and then dephosphorylated by the C phosphatase upon transition to the biofilm state, leading to induction of Pf4. But then why doesn't Pf4 get induced in PAO1 cells that don't have the A/B kinases (or the phosphatase)? This facet of the story still doesn't make sense to me unless I missed something.

MvaU isn't the only target of the kinases. I don't think this facet of the story is adequately addressed still.

Additionally, only a fraction of MvaU gets phosphorylated - I had asked about this last time and it was addressed in the responses where the authors indicate 0.5-20% gets phosphorylated, but I don't see it in the revised paper.

I'm confused why Gp59 isn't mutated in every escape phage (Fig. 6c) if it's able to activate KKP when they're co-expressed?

The data in Fig. 6f are not very convincing. There is an extremely modest change in phosphatase activity and it's not clear whether such a modest effect is responsible in vivo. The authors argue that "since PfpC lies at the top of the KKP signaling cascade, even relatively minor effects on phosphatase activity could feasibly result in substantial impacts on phosphorylation outcomes." That's plausible, but not at all proven by the data presented.

Gp59 and the phosphatase co-IP when coexpressed. Do they also interact, or at least co-IP, during a phage infection?

Reviewer #2 (Remarks to the Author):

We thank the authors for a great response to our comments. The paper really improved. We have no further comments. Congratulations on this exciting piece of science !

REVIEWER COMMENTS

Reviewer #1 (Remarks to the Author):

The authors have responded to the initial issues raised and improved the manuscript. There are, however, remaining issues and concerns. The key concerns can perhaps best be summarized by highlighting two problematic passages of the text, one in the Abstract and one in the Discussion: In the abstract the authors write: "The balance between kinase and phosphatase expression regulates phage production by controlling the phosphorylation of a host nucleoid binding protein." As discussed below, it is not rigorously established that it is the balance in expression that is ultimately responsible for changing the phosphorylation of MvaU, let alone other kinase targets. The expression ratio absolutely does change but whether it is *responsible* is simply not shown. This sort of imprecision in logic and writing really holds the paper back. Additionally, as written, this sentence in the abstract also implies that MvaU is the only or the primary target, which is also not rigorously established. Indeed, the phosphoproteomics suggests multiple targets, most of which haven't been examined in depth. Then, in the Discussion, the authors prematurely conclude/summarize that: "Phosphorylation of a host H-NS protein (MvaU) by PfkA/B represses expression of Pf genes required for prophage activation, keeping Pf6 and Pf4 gene expression largely silent in planktonic cells. However, in biofilms, expression of the entire KKP cassette, including PfpC, alleviates Pf gene repression, presumably via dephosphorylation of MvaU." The last sentence, in particular, is not firmly established yet - the authors do say 'presumably' via dephosphorylation of MvaU, but this really should be established (or not, as it may turn out). And this last quoted sentence also brushes under the rug the issue regarding how kinase/phosphatase balance changes and whether it's a change in expression of kinase, phosphatase, or both, or whether there are other factors involved.

Response: We thank the reviewer for their helpful comments and suggestions to improve this manuscript. We agree with the reviewers' remarks that our data do not fully enable resolution of the complete pathway linking expression levels of KKP components and resulting functional outcomes (i.e., MvaU phosphorylation and prophage induction). As such, we have toned down the text of the Abstract (lines 29-34) and Discussion (lines 320-326) to more clearly communicate that this is a *proposed* model of KKP action and to outline future directions and open questions. We also provide responses to each comment below, noting any associated changes to the revised manuscript.

My primary concerns and a few other minor ones are further described below:

line 38: "provides defense against lytic phages". Against some lytic phages. As written, it implies defense against all lytic phages. Again, this lack of precision in language was problematic in the first version of the paper and continues to plague this one.

Response: We adjusted the wording to "some lytic phages" as suggested. Please see line 35.

The titer of the MPAO1 WT is similar to that of the pf4 and pf6 single deletions. But what phages are produced by the WT? It could be a mix of both, but there could also be one that dominates. It should

be straightforward to assess this and report it. This is partially addressed in the responses to reviewers document, but I don't see a clear explanation or the relevant data in the revised paper.

Response: Thank you for this comment. In **Supplementary Fig. 3cde**, we developed a method that uses Pf4-specific gene (*phiA*) and Pf6-specific gene (*intF6*) primer sets to quantitatively link Ct values from qPCR analyses with Pf-specific PFU values. We used this method to generate the data graphed in **Fig. 2d**, which shows the specific Pf4 and Pf6 phage titers over time in WT MPAO1 biofilms. The data in this panel show that both Pf4 and Pf6 are produced in roughly equivalent amounts over time, and that production of both phages is markedly elevated in the absence of KKP. To clarify to readers that Pf4 and Pf6 are co-produced, we added a similar quantification of Pf4 and Pf6 levels in WT MPAO1 to the **revised Fig. 1f**.

* lines 127-139: This is a major issue. As written, this section concludes that the change in expression of the phosphatase relative to the kinases is *responsible* for the induction of prophages in biofilms. While the expression ratio does change and a null mutant of kinase A shows a clear phenotype, these data alone do not allow one to conclude that it is the change in RNA levels that is responsible. To make that conclusion, the authors need to show that a mutation or genetic change preventing only the change in expression ratio has the same phenotype as the null mutant.

Response: The reviewer raises a valid point that our data do not exclude potential functional mechanisms for KKP that are not directly related to changes in the component expression ratios. Nevertheless, we note that in addition to the bulk transcriptional analyses and gene deletion strains, some of our data addresses the specific question of manipulation of gene expression levels and their impact(s) on KKP output. In particular, we observed that intentionally skewing the expression ratio by overexpressing *pfkA* and *pfkB* in MPAO1 led to an increase in MvaU phosphorylation (**Fig. 2f**, right panel). This was the predicted outcome, since PfpC was not similarly induced in this system. Restoration of stoichiometric gene content by including *pfpC* in the induction vector returned MvaU phosphorylation to normal levels (**Fig. 2f**, right panel), demonstrating that manipulation of gene expression/ratios alone should be sufficient to control KKP output. We agree that the experiment suggested by the reviewer would be another way to link expression ratios to phenotypes, perhaps at a higher resolution than the gene deletions we used in the current study. However, since we do not yet know the specific molecular trigger(s) of the expression shifts and do not have a readily identifiable mutation that would prevent only expression changes, we believe this work would be beyond the scope of the current study. We have incorporated this idea into the text to reflect this future direction: "In particular, the precise environmental factors that trigger differential expression of KKP in planktonic and biofilm cells, which may include factors such as quorum sensing molecules and c-di-GMP, remain to be investigated. Identification of such triggers should enable finer-scale manipulation of KKP expression ratios to determine their precise relationship to KKP signaling outcomes." on **lines 370-373**.

* Related to the point above: it remains unclear what triggers the KKP-dependent induction of Pf4 in biofilms. In other words, what causes the change in expression ratio, if that's indeed what's responsible for the induction? Or, if that's not what ultimately drives induction, what is responsible? (It should also be noted that in Fig. 2b it's not just the ratio of phosphatase to kinase that changes in

biofilms, but the relative expression of the kinases. Moreover, it must be emphasized that these are RNA levels - we don't know what's happening at the protein level at all.)

Response: We thank the reviewer for this interesting comment, which was also raised by reviewer two in the initial round of review. Since biofilm formation is a complicated and dynamic process, many factors likely coordinately control KKP-dependent prophage induction in these conditions. While we consider definition of these triggers beyond the scope of the current study, we include here preliminary data from our ongoing work for the reviewer's interest. Notably, manipulation of oxidative stress-related responses in MPAO1 strongly influence Pf induction (**Fig. R1A**). We have also observed that deletion of genes controlling c-di-GMP synthesis and quorum sensing also influence production of Pf phages in biofilms (**Fig. R1B**). For at least two of these genes (*wspF* and *siaD*), we have observed changes in KKP component expression ratios, an exciting result for further investigation. We are keenly interested in continuing to decipher how environmental and genetic factors may interact in the control of prophage induction during *P. aeruginosa* biofilm formation. We included a discussion of this subject (as mentioned in the previous reply) on **lines 370-373**. We agree with the reviewer that the data in **Fig.2b** show the relative expression of the kinases and phosphatase genes and not proteins. We have emphasized this distinction, including the idea that the kinase expression ratios themselves may contribute to KKP function, on **lines 121-131**.

Fig. R1. The production of Pf phages in MPAO1 biofilms is controlled by oxidative stress, c-di-GMP and quorum sensing. (A) The production of Pf phages in MPAO1 biofilms can be inhibited by reducing oxidative stress with the addition of antioxidant N-acetylcysteine (NAC). Deletion of the oxidative stress-related regulator gene *oxyR* also inhibited Pf phage production. **(B)**, The production of Pf phages in MPAO1 biofilms may depend on intact c-di-GMP and quorum sensing. Deletion of the c-di-GMP-related *wspF* gene results in higher c-di-GMP levels in cells, while deletion of *siaD* gene results in lower c-di-GMP levels in cells. *pqsR*, *rhIR* and *lasR* have functions in *P. aeruginosa* quorum sensing.

The model in Fig. 2h posits that MvaU is phosphorylated by the A/B kinases and then dephosphorylated by the C phosphatase upon transition to the biofilm state, leading to induction of Pf4. But then why doesn't Pf4 get induced in PAO1 cells that don't have the A/B kinases (or the phosphatase)? This facet of the story still doesn't make sense to me unless I missed something.

Response: We thank the reviewer for the opportunity to clarify this point, which relates to our ongoing long-term studies on the multi-layered complexity of genetic control of Pf production in *P. aeruginosa*. In this case, the answer to this apparent paradox lies in the absence of a functional Pf receptor in PAO1, which blocks phage re-infection downstream of prophage induction, limiting overall Pf4 production. Restoration of a functional Pf receptor enabled robust Pf4 production in PAO1 biofilms. Critically, this phenotype was silenced by subsequent introduction of the KKP cassette from MPAO1 into this engineered PAO1 background, suggesting that KKP and receptor status represent two distinct facets of the ongoing arms race between *P. aeruginosa* and its prophages. Although we included an explanation of this phenomenon in earlier versions of the manuscript and discussed this subject in the initial rebuttal, given the manuscript's focus on KKP's dual functions in prophage regulation and phage defense, we elected to turn this data into an independent work that further expands on Pf regulation and superinfection in *P. aeruginosa*. We explain this mechanism in detail for the reviewer in the following paragraph.

Beyond their lack of KKP, we noted that PAO1 *P. aeruginosa* also harbor a 4 nt insertion in the type IV pilus (T4P) component protein encoded by *pilC* (5'-ACTG-3' insertion 78 nt downstream of the start codon). This insertion causes a frameshift mutation after the 28th residue of PilC, rendering it non-functional in PAO1. The T4P is known to be the host receptor for Pf phages ¹. When we replaced the non-functional *pilC* allele in PAO1 with *pilC* from MPAO1, we observed that these PAO1 Δ *pilC*_{PAO1}::*pilC*_{MPAO1} cells now produced large quantities of Pf phages capable of superinfection (Fig. R2A). This phenotype closely resembled that of the Δ *pfkA* MPAO1 mutant (Fig. 2d), consistent with the idea that the absence of KKP combined with the presence of an intact receptor enables unfettered phage production. Heterologous expression of KKP from MPAO1 by PAO1 Δ *pilC*_{PAO1}::*pilC*_{MPAO1}::KKP_{MP} then tamped down phage production as expected (Fig. R2A). Importantly, this phenotype also functioned in reverse – disruption of re-infection by replacing MPAO1's *pilC* with the non-functional PAO1 *pilC* allele abolished Pf phage production in biofilms (Fig. R2B). We interpret these data as indicating that robust Pf4 production in biofilms involves not only Pf4 prophage excision, but also Pf4 re-infection that is T4P-dependent. The loss-of-function mutation of *pilC* in PAO1 (which blocks Pf4 re-infection) and the acquisition of Pf6-encoded KKP in MPAO1 (inhibits Pf4 excision and replication) likely reflect the complicated arm race between Pf phages. These observations also reflect the continuing divergence of the phage biology of the PAO1 and MPAO1 *P. aeruginosa* sublines since their separation in 1955 from the Holloway Lab.

Fig. R2. PilC mutation in PAO1 hindered the production of superinfective Pf in biofilm.

(A) Pf4 phage titers in effluents of PAO1, PAO1 Δ *pilC*_{PAO1}::*pilC*_{MPAO1},

PAO1 Δ *pilC*_{PAO1}::*pilC*_{MPAO1}::KKP_{MP} biofilms at indicated days on PAO1 Δ *pilC*_{PAO1}::*pilC*_{MPAO1} lawns ; (B)

Pf4 and Pf6 phage titers in effluents of MPAO1 and PAO1 Δ *pilC*_{MPAO1::pilC}_{PAO1} biofilms at indicated days on MPAO1 lawns.

MvaU isn't the only target of the kinases. I don't think this facet of the story is adequately addressed still.

Response: The reviewer is correct that other targets of the kinases PfkA/PfkB were identified by phosphoproteomics. We note that the phosphomimetic version of MvaU (MvaU^{S67D}) had a potent repressive effect on Pf production in biofilms, consistent with our model where MvaU is a critical mediator of KKP-dependent control of Pf prophage induction. In this study, we focused on MvaU as we had previously shown this protein to inhibit the production of Pf4 phages, but its regulation was not known². We plan to screen other identified kinase targets for their involvement in the production of Pf phages in future studies and have more explicitly explained this point in the Discussion on lines 320-322, as well as mentioned additional potential targets of KKP on lines 138-140.

Additionally, only a fraction of MvaU gets phosphorylated - I had asked about this last time and it was addressed in the responses where the authors indicate 0.5-20% gets phosphorylated, but I don't see it in the revised paper.

Response: We apologize for this oversight and have added this data to the revised paper on lines 151-152.

I'm confused why Gp59 isn't mutated in every escape phage (Fig. 6c) if it's able to activate KKP when they're co-expressed?

Response: We apologize for the confusion on this point. In **Fig. 6c**, we sequenced evolved T4 phage pools (the 10th passage), which contained both escaped and non-escaped phages. Therefore, we did not expect to observe Gp59 mutations in every phage. In this experimental workflow, we next isolated single escape phages, and as shown in **Fig. 6d**, all three isolated and validated escape phages contained the Gp59^{G32E} mutation.

The data in Fig. 6f are not very convincing. There is an extremely modest change in phosphatase activity and it's not clear whether such a modest effect is responsible *in vivo*. The authors argue that "since PfpC lies at the top of the KKP signaling cascade, even relatively minor effects on phosphatase activity could feasibly result in substantial impacts on phosphorylation outcomes." That's plausible, but not at all proven by the data presented.

Response: We agree that the *in vitro* inhibition of PfpC by Gp59 was modest and now explicitly state this in line 297. However, the true level of inhibition may be higher *in vivo* in the physiological environment where the interaction actually occurs. For example, an additional host cofactor may be required for efficient binding and inhibition of PfpC by Gp59, or the biochemical conditions where we tested phosphatase activity do not fully reflect the conditions that PfpC would experience in the cell. We modified our language around this claim to reflect this idea (lines 341-347): "Although the *in vitro* inhibition of PfpC_{EC039} by Gp59 was modest, since PfpC lies at the top of the KKP signaling

cascade, it is plausible that even relatively minor effects on phosphatase activity could feasibly result in the substantial impacts on downstream phosphorylation of host substrates we observed. Additionally, the true level of PfpC inhibition may be higher *in vivo*, as our data do not account for the possible contributions of additional host co-factors that may promote Gp59 binding to PfpC, or specific physiological biochemical conditions that PfpC may experience in the cell.”

Gp59 and the phosphatase co-IP when coexpressed. Do they also interact, or at least co-IP, during a phage infection?

Response: As the reviewer suggested, we attempted to use His-PfpC_{EC039} protein to pull down Gp59 during T4 infection, but were not successful at verifying the interaction through this method using our standard protocols. We think this may be due to very low expression levels of Gp59, perhaps necessitating more sensitive mass spectrometry-based detection techniques. In transcriptomic data we have collected from T4-infected K-12 *E. coli*, *gp59* transcripts are 30x less abundant (FPKM = 107) than the average T4 gene (FPKM: 2956), suggesting that Gp59 is a low abundance protein that is not readily pulled out of complex infection lysates.

Reviewer #2 (Remarks to the Author):

We thank the authors for a great response to our comments. The paper really improved. We have no further comments. Congratulations on this exciting piece of science !

Response: We thank the reviewer for their suggestions, which markedly improved the revised manuscript.

References:

- 1 Wang, W. *et al.* Filamentous prophage capsid proteins contribute to superinfection exclusion and phage defence in *Pseudomonas aeruginosa*. *Environ. Microbiol.* **24**, 4285–4298 (2022).
- 2 Li, Y. *et al.* Excisionase in Pf filamentous prophage controls lysis-lysogeny decision-making in *Pseudomonas aeruginosa*. *Mol. Microbiol.* **111**, 495–513 (2019).